# DOCTOR: A Simple Method for Detecting Misclassification Errors

**Federica Granese**[*][†]
Lix, Inria, Institute Polytechnique de Paris,
Sapienza University of Rome
federica.granese@inria.fr

**Marco Romanelli**[*]
L2S, CentraleSupélec,
CNRS, Université Paris Saclay
marco.romanelli@centralesupelec.fr

**Daniele Gorla**
Sapienza University of Rome
gorla@di.uniroma1.it

**Catuscia Palamidessi**[†]
Lix, Inria, Institute Polytechnique de Paris,
catuscia@lix.polytechnique.fr

**Pablo Piantanida**[‡]
L2S, CentraleSupélec,
CNRS, Université Paris Saclay
pablo.piantanida@centralesupelec.fr

## Abstract

Deep neural networks (DNNs) have shown to perform very well on large scale object recognition problems and lead to widespread use for real-world applications, including situations where DNN are implemented as "black boxes". A promising approach to secure their use is to accept decisions that are likely to be correct while discarding the others. In this work, we propose DOCTOR, a simple method that aims to identify whether the prediction of a DNN classifier should (or should not) be trusted so that, consequently, it would be possible to accept it or to reject it. Two scenarios are investigated: Totally Black Box (TBB) where only the soft-predictions are available and Partially Black Box (PBB) where gradient-propagation to perform input pre-processing is allowed. Empirically, we show that DOCTOR outperforms all state-of-the-art methods on various well-known images and sentiment analysis datasets. In particular, we observe a reduction of up to $4\%$ of the false rejection rate (FRR) in the PBB scenario. DOCTOR can be applied to any pre-trained model, it does not require prior information about the underlying dataset and is as simple as the simplest available methods in the literature.

## 1   Introduction

With the advancement of state-of-the-art Deep Neural Networks (DNNs), there has been rapid adoption of these technologies in a broad range of applications to critical systems, such as autonomous driving vehicles or industrial robots, including–but not limited to–classification and decision making tasks. Nevertheless, these solutions still exhibit unwanted behaviors as they tend to be overconfident even in presence of wrong decisions [17]. Developing methods and tools to make these algorithms

---

[*]These authors contributed equally to this work

[†]This paper is supported by the ERC project Hypatia under the European Unions Horizon 2020 research and innovation program. Grant agreement №835294.

[‡]This project has received funding from the European Union's Horizon 2020 research and innovation programme under the Marie Skłodowska-Curie grant agreement №792464.

reliable, in particular for non-specialists who may treat them as "black boxes" with no further checks, constitutes a core challenge. Recently, the study of safety AI methods has gained ground, and many efforts have been made in several areas [7, 8, 9, 11, 26, 35, 36]. In this paper, we investigate a simple method capable of detecting whether a prediction of a classifier is likely to be correct, and therefore should be trusted, or it is not, and should be rejected.

Deep learning pursues the idea of learning effective representations from the data itself by training with the implicit assumption that the test data distribution should be similar to the training data distribution. However, when applied to real-world tasks, this assumption does not hold true, leading to a significant increase of misclassification errors. Although classic approaches to Out-Of-Distribution (OOD) detection [1, 12, 21, 23, 33] are not directly concerned with detecting misclassification errors, they are intended to prevent those errors indirectly by identifying potential drifts of the testing distribution. What the above ODD methods have in common with our work is that samples drawn from the in-distribution are more likely to be correctly classified than those from a different distribution. Indeed, the model's soft-predictions for in-distribution samples tend to be generally peaky in correspondence to the correct class label while they tend to be less peaky for input samples drawn from a different distribution [12]. In general, most of these works consider *white-box* scenarios, where the hidden layers of the architecture are accessible or the corresponding weights are tuned during the training phase. A very effective approach to OOD detection is ODIN [23] which involves the use of temperature scaling and the addition of small perturbations to input samples. A related solution is introduced in [6] where the maximum soft-probability is called *softmax response*. Within this approach, the softmax response decides whether the classifier is confident enough in its prediction or not. A different approach to OOD detection is given by the use of the Mahalanobis distance [14, 22], which consists in calculating how much the observed out-distribution sample deviate from the in-distribution ones but assuming the latter are given.

## 1.1 Summary of contributions

Our work tackles the problem of identifying whether the prediction of a classifier should or should not be trusted, no matter if they are made on out or in-distribution samples, and advances the state-of-the-art in multiple ways.

- From the theoretical point of view, we derive the trade-off between two types of error probabilities: Type-I, that refers to the rejection of the classification for an input that would be correctly classified, and Type-II, that refers to the acceptance of the classification for an input that would not be correctly classified (Proposition 3.1). The characterization of the optimal discriminator in eq. (10) allows us to devise a feasible implementation of it, based on the softmax probability (Proposition 3.2).

- From the algorithmic point of view, inspired by our theoretical analysis, we propose DOCTOR a new discriminator (Definition 2), which yields a simple and flexible framework to detect whether a decision made by a model is likely to be correct or not. We distinguish two scenarios under which DOCTOR can be deployed: Totally Black Box (TBB) where only the soft-predictions are available, hence gradient-propagation to perform input pre-processing is not allowed, and Partially Black Box (PBB) where we further allow method-specific inputs perturbations.

- From the experimental point of view, we show that DOCTOR outperforms comparable state-of-the-art methods (e.g., ODIN [23], softmax response [6] and Mahalanobis distance [22]) on datasets including both in-distribution and out-of-distribution samples, and different architectures with various expressibilities, under both TBB and PBB. A key ingredient of DOCTOR is to fully exploit all available information contained in the soft-probabilities of the predictions (not only their maximum).

## 1.2 Related works

Recent works have shown that the accuracy of a classifier and its ability to output soft-predictions that represent the true posteriors estimate can be totally disjointed [9, 19, 20]. Furthermore, models often tend to be overconfident about their decision even when their predictions fail [11, 17]. This motivates a novel research area that strives to develop methods to assess when decisions made by classifiers should or should not be trusted. Although the detection of OOD samples is a different

(domain) problem, it is naturally expected that samples from a distribution that is significantly different from the training one cannot be correctly classified. In [23], the authors propose a method which increases the peakiness of the softmax output by perturbing the input samples and applying temperature scaling [9, 13, 29] to the classifier logits in order to better detect in-distribution samples. It is worth noticing that this method requires additional information on the internal structure of the latent code of the model. A very different approach [14, 22] tackles OOD detection by using the *Mahalanobis distance*. Although this approach appears to be more powerful, it also requires additional samples to learn the mean by class and the covariance matrix of the in-distribution. In [4], classifiers are trained to output calibrated confidence estimates that are used to perform OOD detection. A related line of research is concerned with the problem of *selective predictions* (aka *reject options*) in deep neural networks. The main motivation for selective prediction is reducing the error rate by abstaining from prediction when in doubt, while keeping the number of correctly classified samples as high as possible [5, 6, 7]. The idea is to combine classifiers with *rejection functions* by observing the classifiers' output, without using any supervision, to decide whether to accept or to reject the classification outcome. In [6], the authors introduce *softmax response*, a rejection function which compares the maximum soft-probability to a pre-determined threshold to decide whether to accept or reject the class prediction given by the model.

## 2 Main Definitions and Preliminaries

### 2.1 Basic definitions

We start by introducing some definitions and background; then, we describe our statistical model and some useful properties about the underlying detection problem. Let $\mathcal{X} \subseteq \mathbb{R}^d$ be the (possibly continuous) feature space and let $\mathcal{Y} = \{1, \ldots, C\}$ denote the concept of the label space related to some task of interest. Moreover, let $p_{XY}$ be the underlying (unknown) probability density function (pdf) over $\mathcal{X} \times \mathcal{Y}$. Let $\mathcal{D}_n = \{(\mathbf{x}_1, y_1), \ldots, (\mathbf{x}_n, y_n)\} \sim p_{XY}$ be a random realization of $n$ i.i.d. samples according to $p_{XY}$ denoting the *training set*, where $\mathbf{x}_i \in \mathcal{X}$ is the input (feature), $y_i \in \mathcal{Y}$ is the output class among $C$ possible classes and $n$ denotes the size of the training set. A predictor $f_{\mathcal{D}_n} : \mathcal{X} \to \mathcal{Y}$ uses the inferred model $P_{\widehat{Y}|X} \equiv P_{\widehat{Y}|X}(y|\mathbf{x}; \mathcal{D}_n)$ based on the training set,

$$f_{\mathcal{D}_n}(\mathbf{x}) \equiv f_n(\mathbf{x}; \mathcal{D}_n) \triangleq \arg\max_{y \in \mathcal{Y}} P_{\widehat{Y}|X}(y|\mathbf{x}; \mathcal{D}_n),$$

and tries to approximate the optimal (Bayes) decision rule $f^\star(\mathbf{x}) \triangleq \arg\max_{y \in \mathcal{Y}} P_{Y|X}(y|\mathbf{x})$. Notice that $P_{\widehat{Y}|X}$ can be interpreted as the prediction of the class (label) posterior probability given a sample (e.g., $P_{\widehat{Y}|X}(y|\mathbf{x}) \equiv \text{softmax}(\mathbf{x})_y$), while $P_{Y|X}$ is the true (unknown) probability. In several practical scenarios $P_{\widehat{Y}|X}$ does not perfectly match $P_{Y|X}$ and still $f_{\mathcal{D}_n} \approx f^\star$ (cf. [9]).

### 2.2 Error variable

Let $E(\mathbf{x}) \triangleq \mathbb{1}\left[Y \neq f_{\mathcal{D}_n}(\mathbf{x})\right]$ denote the error variable for a given $\mathbf{x} \in \mathcal{X}$ corresponding to $f_{\mathcal{D}_n}$, i.e., where we denote with $\mathbb{1}[\mathcal{E}]$ the indicator vector which outputs 1 if the event $\mathcal{E}$ is true and 0 otherwise. Similarly, we can define the self-error variable $\widehat{E}(\mathbf{x}) \triangleq \mathbb{1}\left[\widehat{Y} \neq f_{\mathcal{D}_n}(\mathbf{x})\right]$ also corresponding to the inferred predictor $f_{\mathcal{D}_n}$ but based on the prediction model $P_{\widehat{Y}|X}$ of the class posterior probability. Notice that $\widehat{E}(\mathbf{x})$ is observable since the underlying distribution is known. However, $E(\mathbf{x})$ cannot be observed and in general these binary variables do not coincide.

At this stage, it is convenient to introduce the notions of *probability of classification error* for a given $\mathbf{x} \in \mathcal{X}$ w.r.t. both the true class posterior and the predicted probabilities:

$$\text{Pe}(\mathbf{x}) \triangleq \mathbb{E}\left[E(\mathbf{x})|\mathbf{x}\right] = 1 - P_{Y|X}\left(f_{\mathcal{D}_n}(\mathbf{x})|\mathbf{x}\right), \tag{1}$$

$$\widehat{\text{Pe}}(\mathbf{x}) \triangleq \mathbb{E}\left[\widehat{E}(\mathbf{x})|\mathbf{x}\right] = 1 - P_{\widehat{Y}|X}\left(f_{\mathcal{D}_n}(\mathbf{x})|\mathbf{x}\right). \tag{2}$$

Notice that $\widehat{\text{Pe}}(\mathbf{x})$ represents the probability of misclassification of the sample $\mathbf{x}$ with respect to the softmax probability $P_{\widehat{Y}|X}$, which can be interpreted as the model's approximation of nature $P_{Y|X}$. Such approximation is close when the model is well-calibrated. Obviously, $\text{Pe}^\star(\mathbf{x}) \leq \text{Pe}(\mathbf{x})$ for all $\mathbf{x} \in \mathcal{X}$, where $\text{Pe}^\star(\mathbf{x})$ corresponds to the minimum error of the Bayes classifier:

$\text{Pe}^\star(\mathbf{x}) = 1 - P_{Y|X}\left(f^\star(\mathbf{x})|\mathbf{x}\right)$. It is worth mentioning that, by averaging (1) over the data distribution, we obtain the error rate of the classifier $f_{\mathcal{D}_n}$. Although $\widehat{\text{Pe}}(\mathbf{x})$ provides a valuable candidate to infer the unknown error variable $E(\mathbf{x})$, it is easy to check that

$$\max\left\{\text{Pe}(\mathbf{x}), \widehat{\text{Pe}}(\mathbf{x})\right\} - \Pr\left(\widehat{Y} = Y|\mathbf{x}\right) \leq \Pr\left\{\widehat{E}(\mathbf{x}) \neq E(\mathbf{x})|\mathbf{x}\right\} \leq \Pr\left(\widehat{Y} \neq Y|\mathbf{x}\right), \quad (3)$$

which in particular implies that the error incurred in using $\widehat{E}(\mathbf{x})$ to predict $E(\mathbf{x})$ is lower bounded by the classification error per sample (1). The proofs are in Supplementary material (Appendix A.3).

In this paper, we aim at identifying a discriminator capable of distinguishing between inputs $\mathbf{x}$ for which we can trust the predictions of the classifier $f_{\mathcal{D}_n}(\mathbf{x})$ (i.e., $E(\mathbf{x}) = 0$) and those for which we should not trust predictions (i.e., $E(\mathbf{x}) = 1$). In the next section, we will show that the function $\text{Pe}(\mathbf{x}) : \mathcal{X} \mapsto [0, 1]$ plays a central role in the characterization of the optimal discriminator. However, $\text{Pe}(\mathbf{x})$ is not available in practical scenarios and the direct estimation (e.g., based on pairs of inputs and labels) of the true class posterior probability $P_{Y|X}$ cannot be performed. Notice that it is not possible to sample the conditional pdf $P_{Y|X}$ for each input $\mathbf{x} \in \mathcal{X}$. As a matter of fact, it is well-known that the application of direct methods for this estimation will lead to ill-posed problems, as shown in [32].

### 2.3 Statistical model for detection

Given a data sample $\mathbf{x} \in \mathcal{X}$ and an unobserved random label $y \in \mathcal{Y}$ drawn from the unknown distribution $p_{XY}$, we wish to predict the realization of the unobserved error variable $E \triangleq \mathbb{1}[Y \neq f_{\mathcal{D}_n}(\mathbf{X})]$. To this end, we will model the data distribution as a mixture pdfs,

$$p_{XY}(\mathbf{x}, y) \equiv P_E(1)p_{XY|E}(\mathbf{x}, y|1) + P_E(0)p_{XY|E}(\mathbf{x}, y|0),$$

where $p_{XY|E}(\mathbf{x}, y|1)$ denotes the pdf truncated to the error event $\{E = 1\}$ (i.e., the hard decision fails) and $p_{XY|E}(\mathbf{x}, y|0)$ is the pdf truncated to the success event $\{E = 0\}$ (i.e., the hard decision succeeds). By taking the marginal of $p_{XY}$ over the labels, we obtain: $p_X(\mathbf{x}) = P_E(1)p_{X|E}(\mathbf{x}|1) + P_E(0)p_{X|E}(\mathbf{x}|0)$. First, observe that the problem at hand is to infer $E$ from $(\mathbf{x}, P_{\widehat{Y}|X})$ since $Y$ is not observed. Second, we further emphasize that in the present framework we assume that there are no available (extra) samples for training a discriminator to distinguish between $p_{X|E}(\mathbf{x}|0)$ and $p_{X|E}(\mathbf{x}|1)$. It is worth mentioning that a well-trained classifier would imply $P_E(1) \ll P_E(0)$, since in that case we should have very few classification errors. However, this also implies that it would be very unlikely to have enough samples available to train a good enough discriminator.

## 3 Performance Metrics and Discriminators

### 3.1 Performance metrics and optimal discriminator

We aim to distinguish between samples for which the predictions cannot be trusted and samples for which predictions should be trusted. We first state the optimal rejection region, given by (4), where we suppose the existence of an oracle who knows all the involved probability distributions.

**Definition 1** (Most powerful discriminator). For any $0 < \gamma < \infty$, define the decision region:

$$\mathcal{A}(\gamma) \triangleq \left\{\mathbf{x} \in \mathcal{X} : p_{X|E}(\mathbf{x}|1) > \gamma \cdot p_{X|E}(\mathbf{x}|0)\right\}. \quad (4)$$

The most powerful (Oracle) discriminator at threshold $\gamma$ is defined by setting $D(\mathbf{x}, \gamma) = 1$ for all $\mathbf{x} \in \mathcal{A}(\gamma)$ for which the prediction is rejected (i.e., $\widehat{E} = 1$) and otherwise $D(\mathbf{x}, \gamma) = 0$ for all $\mathbf{x} \notin \mathcal{A}(\gamma)$ for which the prediction is accepted.

In Proposition 3.1, we establish the characterization of the fundamental performance of the most powerful (Oracle) discriminator by providing a lower bound on the error achieved by any discriminator and show that this bound is achievable by setting $\gamma = 1$. Furthermore, we connect this result to the Bayesian error rate of this optimal discriminator.

**Proposition 3.1** (Performance of the discriminator). For any given decision region $\mathcal{A} \subset \mathcal{X}$, let

$$\epsilon_0(\mathcal{A}) \triangleq \int_{\mathcal{A}} p_{X|E}(\mathbf{x}|0)d\mathbf{x}, \quad \text{and} \quad \epsilon_1(\mathcal{A}^c) \triangleq \int_{\mathcal{A}^c} p_{X|E}(\mathbf{x}|1)d\mathbf{x}, \quad (5)$$

be the Type-I (rejection of the class prediction of an input $\mathbf{x}$ that would be correctly classified) and Type-II (acceptance of the class prediction of an input $\mathbf{x}$ that would not be correctly classified) error probability, respectively. Then,

$$\epsilon_0(\mathcal{A}) + \epsilon_1(\mathcal{A}^c) \geq 1 - \left\| p_{X|E=1} - p_{X|E=0} \right\|_{\text{TV}} \tag{6}$$

$$= 1 - \frac{1}{2} \int_{\mathcal{X}} |p_{X|E=1}(\mathbf{x}) - p_{X|E=0}(\mathbf{x})| d\mathbf{x}. \tag{7}$$

Equality is achieved by choosing the optimal decision region $\mathcal{A}^\star \equiv \mathcal{A}(1)$ in Definition 1. If the hypotheses are equally distributed, the minimum Bayesian error satisfies:

$$2 \Pr\{D(\mathbf{X}) \neq E(\mathbf{X})\} \geq 1 - \left\| p_{X|E=1} - p_{X|E=0} \right\|_{\text{TV}}. \tag{8}$$

Equality is achieved by using the optimal decision region.

Expressions (7) and (8) provide lower bounds for the total error of an arbitrary discriminator. The proof of this proposition is relegated to the Supplementary material (Appendix A.1). Using Bayes we can rewrite (4) via the posteriors as:

$$\mathcal{A}(\gamma) = \left\{ \mathbf{x} \in \mathcal{X} : P_{E|X}(1|\mathbf{x}) P_E(0) > \gamma \cdot \left(1 - P_{E|X}(1|\mathbf{x})\right) P_E(1) \right\}. \tag{9}$$

From (9), it is easy to check that $P_{E|X}(1|\mathbf{x}) = 1 - P_{Y|X}(f_{\mathcal{D}_n}(\mathbf{x})|\mathbf{x}) = \text{Pe}(\mathbf{x})$, and hence, the decision region $\mathcal{A}(\gamma)$ can be reformulated as:

$$\mathcal{A}(\gamma') = \left\{ \mathbf{x} \in \mathcal{X} : \frac{\text{Pe}(\mathbf{x})}{1 - \text{Pe}(\mathbf{x})} > \gamma' \right\} = \left\{ \mathbf{x} \in \mathcal{X} : \text{Pe}(\mathbf{x}) > \frac{\gamma'}{(\gamma' + 1)} \right\}, \tag{10}$$

where $\gamma' \triangleq \gamma \cdot \frac{P_E(1)}{P_E(0)}$ and $0 < \gamma' < \infty$. According to (10) and Proposition (3.1), the optimal discriminator is given by $D^\star(\mathbf{x}, \gamma') = 1$, whenever $\mathbf{x} \in \mathcal{A}(\gamma')$, and $D^\star(\mathbf{x}, \gamma') = 0$, otherwise. The main difficulty arises here since the error probability function of an input: $\mathbf{x} \mapsto \text{Pe}(\mathbf{x})$ is not known and in general cannot be learned from training samples.

## 3.2 DOCTOR discriminator

We start by deriving an approximation to the unknown function $\mathbf{x} \mapsto \text{Pe}(\mathbf{x})$ which can be used to devise the decision region in expression (10). First, we state the following:

**Proposition 3.2.** Let $\widehat{\mathrm{g}}(\mathbf{x})$ be defined by

$$1 - \widehat{\mathrm{g}}(\mathbf{x}) \triangleq \sum_{y \in \mathcal{Y}} P_{\widehat{Y}|X}(y|\mathbf{x}) \Pr\left(\widehat{Y} \neq y|\mathbf{x}\right) = 1 - \sum_{y \in \mathcal{Y}} P_{\widehat{Y}|X}^2(y|\mathbf{x}), \tag{11}$$

for each $\mathbf{x} \in \mathcal{X}$, which indicates the probability of incorrectly classifying a feature $\mathbf{x}$ if it was randomly labeled according to the model distribution $P_{\widehat{Y}|X}$ trained based on the dataset. Then,

$$(1 - \sqrt{\widehat{\mathrm{g}}(\mathbf{x})}) - \Delta(\mathbf{x}) \leq \text{Pe}(\mathbf{x}) \leq (1 - \widehat{\mathrm{g}}(\mathbf{x})) + \Delta(\mathbf{x}), \tag{12}$$

where $\Delta(\mathbf{x}) \triangleq 2\sqrt{2 \, \text{KL}\left(P_{Y|X}(\cdot|\mathbf{x}) \| P_{\widehat{Y}|X}(\cdot|\mathbf{x})\right)}$ and denotes the Kullback–Leibler (KL) divergence (further details are provided in Supplementary material Appendix A.2).

## 3.3 Discussion

It is worth emphasizing that expressions in (12) provide bounds to the unknown function $\mathbf{x} \mapsto \text{Pe}(\mathbf{x})$ using a known statistics $\mathbf{x} \mapsto 1 - \widehat{\mathrm{g}}(\mathbf{x})$, which is based on the soft-probability of the predictor. On the other hand, $0 \leq \widehat{\mathrm{g}}(\mathbf{x}) \leq \sqrt{\widehat{\mathrm{g}}(\mathbf{x})} \leq 1$, for all $\mathbf{x} \in \mathcal{X}$, which simply follows using the subadditive of the function $t \mapsto \sqrt{t}$ and the definition of $\widehat{\mathrm{g}}(\mathbf{x})$. By Markov's inequality,

$$\Pr\left(\Delta(\mathbf{X}) \geq \varepsilon(\eta)\right) \leq \eta \quad \text{with} \quad \varepsilon(\eta) = 2\sqrt{2\mathbb{E}_{\mathbf{X}Y}\left[-\log P_{\widehat{Y}|X}(Y|\mathbf{X})\right]/\eta}, \tag{13}$$

for any $\eta > 0$, where $\mathbb{E}_{\mathbf{X}Y}\left[-\log P_{\widehat{Y}|X}(Y|\mathbf{X})\right]$ in (13) is the cross-entropy risk. The latter is expected to be small provided that the model generalizes well. Thus, $\varepsilon(\eta)$ can be expected to be small for a desired confidence $\eta > 0$. Interestingly, (11) turns out to be related to the uncertainty of the classifier via the quadratic Rényi entropy [31]: $-\log_2\left(\widehat{\mathrm{g}}(\mathbf{x})\right) = 2H_2(\widehat{Y}|\mathbf{x}) \leq 2H(\widehat{Y}|\mathbf{x})$, where the latter is the Shannon entropy, i.e., the self-uncertainty of the classifier.

### 3.4 From the theory to a practical discriminator

Our previous discussion suggests that $\widehat{\text{Pe}}(\mathbf{x})$ in (2) may be a valuable candidate to approximate $\text{Pe}(\mathbf{x})$ in the definition of the optimal discriminator (10). On the other hand, Proposition 3.2 suggests that $1 - \widehat{\text{g}}(\mathbf{x})$ can also be a valuable candidate yielding another discriminator. These discriminators, referred to as DOCTOR, are introduced below.

**Definition 2** (DOCTOR). For any $0 < \gamma < \infty$ and $\mathbf{x} \in \mathcal{X}$, define the following discriminators:

$$D_\alpha(\mathbf{x}, \gamma) \triangleq \mathbb{1}\left[1 - \widehat{\text{g}}(\mathbf{x}) > \gamma \cdot \widehat{\text{g}}(\mathbf{x})\right], \qquad D_\beta(\mathbf{x}, \gamma) \triangleq \mathbb{1}\left[\widehat{\text{Pe}}(\mathbf{x}) > \gamma \cdot (1 - \widehat{\text{Pe}}(\mathbf{x}))\right]. \tag{14}$$

Notice that because of Definition 2 and (11), $D_\alpha(\mathbf{x}, \gamma) = \mathbb{1}[1 - \sum_{y \in \mathcal{Y}} \text{softmax}^2(\mathbf{x})_y > \gamma \cdot \sum_{y \in \mathcal{Y}} \text{softmax}^2(\mathbf{x})_y]$; similarly because of Definition 2 and eq. (2), $D_\beta(\mathbf{x}, \gamma) = \mathbb{1}[1 - \max_{y \in \mathcal{Y}} \text{softmax}(\mathbf{x})_y > \gamma \cdot \max_{y \in \mathcal{Y}} \text{softmax}(\mathbf{x})_y]$. The performance of these discriminators will be investigated and compared to state-of-the-art methods in the next section. In the supplementary material (Appendix B), we illustrate how DOCTOR and the optimal discriminator (Definition 1) work on a synthetic data model that is a mixture of two spherical Gaussians with one component per class.

## 4 Experimental Results

In this section we present a collection of experimental results to investigate the effectiveness of DOCTOR, by applying it to several benchmark datasets. We provide publicly available code[1] to reproduce our results, and we give further details on the environment, the parameter setting and the experimental setup in the Supplementary material (Appendix C). We propose a comparison with state-of-the-art methods using similar information. Though we are not concerned with the OOD detection problem, we are confident it is appropriate to compare DOCTOR to methods which use soft-probabilities or at most the output of the latent code, e.g., ODIN [23], softmax response (SR) [6] and Mahalanobis distance (MHLNB) [22]. Since we are focusing on misclassification detection, it is expected that OOD samples should be also detected as classification errors.

**Totally Black Box (TBB) and Partially Black Box (PBB).** We address two different scenarios with respect to the available information about the network. In the TBB only the output of the last layer of the network is available, hence gradient-propagation to perform input pre-processing is not allowed. In the PBB we allow method-specific inputs perturbations. When considering DOCTOR in PBB, for each testing sample $\mathbf{x}$, we calculate the pre-processed sample $\widetilde{\mathbf{x}}$ by adding a small perturbation:

$$\widetilde{\mathbf{x}}^\alpha = \mathbf{x} - \epsilon \times \text{sign}\left[-\nabla_\mathbf{x} \log\left(\frac{1 - \widehat{\text{g}}(\mathbf{x})}{\widehat{\text{g}}(\mathbf{x})}\right)\right], \text{ and } \widetilde{\mathbf{x}}^\beta = \mathbf{x} - \epsilon \times \text{sign}\left[-\nabla_\mathbf{x} \log\left(\frac{\widehat{\text{Pe}}(\mathbf{x})}{1 - \widehat{\text{Pe}}(\mathbf{x})}\right)\right].$$

We will write directly $\widetilde{\mathbf{x}}$ when it is clear from the context which input pre-processing we are referring to. In Supplementary material (Appendix C.2) we further analyze the equations above. When ODIN or MHLNB are used, we pre-process the inputs as in [23] and in [22], respectively.

### 4.1 Review of related methods

**PBB.** We compare DOCTOR (using input pre-processing and temperature scaling) with ODIN and MHLNB. ODIN [23] comprises temperature scaling and input pre-processing via perturbation. Temperature scaling is applied to its scoring function, which has $f_i(\widetilde{\mathbf{x}})$ for the logit of the $i$-th class. Formally, given an input sample $\mathbf{x}$:

$$\text{SODIN}(\widetilde{\mathbf{x}}) = \max_{i=[1:C]} \frac{\exp(f_i(\widetilde{\mathbf{x}})/T)}{\sum_{j=1}^{C} \exp(f_j(\widetilde{\mathbf{x}})/T)}, \ \text{ODIN}(\widetilde{\mathbf{x}}; \delta, T, \epsilon) = \begin{cases} \text{out}, & \text{if } \text{SODIN}(\widetilde{\mathbf{x}}) \leq \delta \\ \text{in}, & \text{if } \text{SODIN}(\widetilde{\mathbf{x}}) > \delta, \end{cases}$$

where $\widetilde{\mathbf{x}}$ represents a magnitude $\epsilon$ perturbation of the original $\mathbf{x}$; $T$ is the temperature scaling parameter; $\delta \in [0, 1]$ is the threshold value; *in* indicates the acceptance decision while *out* indicates the rejection decision. Notice, however, $\gamma$ in DOCTOR and $\delta$ in ODIN, respectively, are defined over two different domains: if $\delta$ denotes a probability, $\gamma$ is a ratio between probabilities. Although ODIN

---

[1]`https://github.com/doctor-public-submission/DOCTOR/`

originally required tuning the hyper-parameter $T$ with out-of-distribution data, it was also shown that a large value for $T$ is generally desirable, suggesting that this gain is achieved at $1000$. Anyway, in this framework, we notice an improvement of ODIN in performance for low values of $T$. Thus we report the best results obtained by ODIN considering the range of hyper-parameters values tested also for DOCTOR (cf. section 4.3). ENERGY [24] comprises the denominator of the softmax activation:

$$\text{ES}(\mathbf{x}) = -T \cdot \log \sum_{j=1}^{C} \exp(f_j(\mathbf{x})/T), \ \ \text{ENERGY}(\mathbf{x}; \xi, T) = \begin{cases} \text{out}, & \text{if } -\text{ES}(\mathbf{x}) \leq \xi \\ \text{in}, & \text{if } -\text{ES}(\mathbf{x}) > \xi, \end{cases}$$

where $\xi \in \mathbb{R}$ is the threshold value. Unlike all the methods considered in this paper, MHLNB [22] requires the knowledge of the training set $\mathcal{D}_n$ which the pre-trained network was trained on to compute its *empirical class mean* $\widehat{\mu}_c$ for each class $c$ and its *empirical covariance* $\widehat{\Sigma}$:

$$\widehat{\mu}_c = \frac{1}{n_c} \sum_{i:\, y_i = c} f(\widetilde{\mathbf{x}}_i); \ \widehat{\Sigma} = \frac{1}{n} \sum_{c \in \mathcal{Y}} \sum_{\forall i:\, y_i = c} (f(\widetilde{\mathbf{x}}_i) - \widehat{\mu}_c)(f(\widetilde{\mathbf{x}}_i) - \widehat{\mu}_c)^{\top},$$

where $n_c$ denotes the number of training samples with label $c$ and $f(\widetilde{\mathbf{x}})$ the logits vector. As MHLNB directly uses the vector of logits, it does not comprise temperature scaling. Given an input sample $\mathbf{x}$:

$$\text{M}(\widetilde{\mathbf{x}}) = \max_{c \in \mathcal{Y}} \ -(f(\widetilde{\mathbf{x}}) - \widehat{\mu}_c)^{\top} \widehat{\Sigma}^{-1} (f(\widetilde{\mathbf{x}}) - \widehat{\mu}_c), \ \ \text{MHLNB}(\widetilde{\mathbf{x}}; \zeta, \epsilon) = \begin{cases} \text{out}, & \text{if } \text{M}(\widetilde{\mathbf{x}}) > \zeta \\ \text{in}, & \text{if } \text{M}(\widetilde{\mathbf{x}}) \leq \zeta, \end{cases}$$

as mentioned above, $\widetilde{\mathbf{x}}$ represents a magnitude $\epsilon$ perturbation of the original $\mathbf{x}$; $\zeta \in \mathbb{R}_+$ is the threshold value; *in* indicates the acceptance decision while *out* indicates the rejection decision.

**TBB.** We compare DOCTOR (without input pre-processing and temperature scaling) with MHLNB (without input pre-processing and with the softmax output layer in place of the logits) and SR. Although both DOCTOR and SR have access to the softmax output of the predictor, a fundamental difference is that, while the former utilizes the softmax output in its entirety, the latter only uses the maximum value, therefore discarding potentially useful information. As it will be shown, this leads to better results for DOCTOR on several datasets (see table 1). We emphasize that, by setting $T = 1$ and $\epsilon = 0$, ODIN reduces to softmax response [6] since $\text{SR}(\mathbf{x}) \equiv \text{SODIN}(\mathbf{x})$.

### 4.2 Detection of misclassification errors, experimental setup and evaluation metrics

Before digging into the detailed discussion of our numerical results, we present an empirical analysis of the behavior of DOCTOR, ODIN, SR and MHLNB when faced with the task of choosing whether to accept or reject the prediction of a given classifier for a certain sample. In Figure 1, we propose a graphical interpretation of the discrimination performance, considering the labeled samples in the dataset TinyImageNet and the ResNet network as the classifier. We separate correctly and incorrectly classified samples according to their true labels in blue and in red, respectively. We remind that the label information is *not* necessary for the discriminators to define acceptance and rejection regions. Then, for each sample we compute the corresponding discriminators' output. These values are binned and reported on the horizontal axis of Figure 1a and Figure 1b for $D_\alpha$, Figure 1c and Figure 1d for $D_\beta$, Figure 1e for SR, Figure 1f for ODIN, Figure 1g and Figure 1h for MHLNB. In each each plot, and according to the corresponding discriminator, the bins' heights represent the frequency of the samples whose value falls within that bin. The intuition is that, if moving along the horizontal axis it is possible to pick a threshold value such that, w.r.t. this value, blue bars are on one side of the plot and red bars on the other, this threshold would correspond to the optimal discriminator, i.e. the discriminator that chooses the optimal acceptance and rejection regions.

In Figure 1g through Figure 1h, we observe that, for MHLNB, no matter how well we choose the threshold value, it is hard to fully separate red and blue bars both in TBB and PBB, i.e. the discriminator fails at defining acceptance and rejection regions so that all the hits can be assigned to the first one and all the mis-classification to the second one. The samples distribution for SR and ODIN in Figure 1e and Figure 1f, respectively, does not look significantly different from the one related to $D_\alpha$ and $D_\beta$ in TBB (Equation (14)). However, the discrimination between samples becomes evident in PBB. This is shown in Figure 1d for $D_\beta$ (eq. (14)) and even more in Figure 1b for $D_\alpha$ (eq. (14)) where, quite clearly, rightly classified samples are clustered on the left-end side of the plot and incorrectly classified samples tend to cluster on the right-end side. This intuition is supported by the results in Table 1.

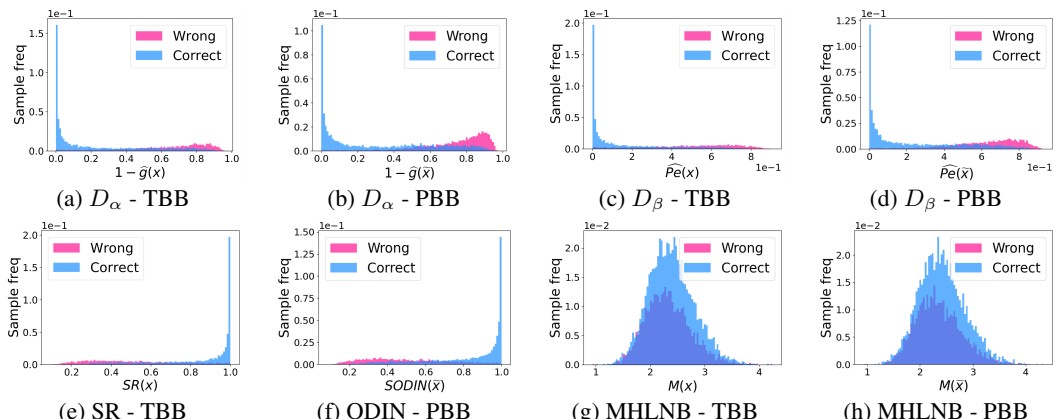

Figure 1: DOCTOR, ODIN, SR and MHLNB to split data samples in TinyImageNet both under TBB and PBB: (a) - (b) show the results for expressions (2); (c) - (d) show the results for (11); (e) shows the results for SR; (f) shows the results for ODIN; (g) - (h) show the results for MHLNB. Histograms for wrongly classified samples (red) and correctly classified samples (blue).

**Datasets and pre-trained networks.** We run experiments on both image and textual datasets. We use CIFAR10 and CIFAR100 [18], TinyImageNet [16] and SVHN [27] as image datasets; IMDb [25], AmazonFashion and AmazonSoftware [28] as textual datasets. Note that, for all the aforementioned datasets, we consider only the test set since we rely on pre-trained models. Along the same lines of [23], we use the pre-trained DenseNet models [15] for CIFAR10, CIFAR100 and SVHN. In addition, we use a pre-trained ResNet model [10] for TinyImageNet, and BERT [3, 34] for the Amazon datasets and IMDb. The accuracy achieved by the aforementioned networks on the test sets is showed in Table 1. According to the invariant properties of the discriminator (see Def. 2) with respect to the soft-probability of the underlying model, permutations of the posterior probabilities vector, due different initialization of the models before the training, do not change the output of Eq. (10), as it is a sum of squared values of the softmax probabilities. This variety of models/datasets characterizes the performance of the proposed method in scenarios with different accuracy levels.

**Evaluation metrics.** We will evaluate the performance according to Proposition (3.1) via the empirical estimates of Type-I and Type-II errors in expressions (5). Throughout this section, when the model's decision for a sample is correct (hit) but is rejected by the discriminator, we refer to such event as *false rejection*; when the model's decision for a sample is not correct (miss) and is rejected by the discriminator, we refer to such event as *true rejection*. Similarly, we refer to a *false acceptance* when a miss is not rejected and to a *true acceptance* when a hit is not rejected. More specifically, let $\mathcal{T}_m = \{(\mathbf{x}_1, y_1), \ldots, (\mathbf{x}_m, y_m)\} \sim p_{XY}$ be the *testing set*, where $\mathbf{x}_i \in \mathcal{X}$ is the input sample, $y_i \in \{1, \ldots, C\}$ is the true class of $\mathbf{x}_i$, and $m$ denotes the size of the testing set. With $j \in \{\alpha, \beta\}$:

$$\mathcal{FR}_j(\gamma) = \{(\mathbf{x}, y) \in \mathcal{T}_m : y = f_{\mathcal{D}_n}(\mathbf{x}), \ D_j(\mathbf{x}, \gamma) = 1\}, \tag{15}$$

$$\mathcal{TR}_j(\gamma) = \{(\mathbf{x}, y) \in \mathcal{T}_m : y \neq f_{\mathcal{D}_n}(\mathbf{x}), \ D_j(\mathbf{x}, \gamma) = 1\}, \tag{16}$$

$$\mathcal{FA}_j(\gamma) = \{(\mathbf{x}, y) \in \mathcal{T}_m : y \neq f_{\mathcal{D}_n}(\mathbf{x}), \ D_j(\mathbf{x}, \gamma) = 0\}, \tag{17}$$

$$\mathcal{TA}_j(\gamma) = \{(\mathbf{x}, y) \in \mathcal{T}_m : y = f_{\mathcal{D}_n}(\mathbf{x}), \ D_j(\mathbf{x}, \gamma) = 0\}. \tag{18}$$

We measure the performance of the test in terms of:

- **FRR** versus **TRR**. The false rejection rate (FRR) represents the probability that a hit is rejected, while the true rejection rate (TRR) is the probability that a miss is rejected.
- **AUROC**. The area under the *Receiver Operating Characteristic curve* (ROC) [2] depicts the relationship between TRR and FRR. The perfect detector corresponds to a score of $100\%$.
- **FRR at 95 % TRR.** This is the probability that a hit is rejected when the TRR is at 95 %.

### 4.3 Experimental results: comparison between different discriminators

**DOCTOR: comparison between $D_\alpha$ and $D_\beta$.** We compare the discriminators $D_\alpha$ and $D_\beta$ introduced in (14) to show how the AUROCs for CIFAR10, CIFAR100, TinyImageNet and SVHN change when

varying the parameters $T$ and $\epsilon$. It is observed that $D_\alpha$ is less sensitive to the selection of $T$: for all the datasets, $D_\alpha$ outperforms $D_\beta$ achieving the best AUROCs by setting $T = 1$. Contrary to $D_\alpha$, $D_\beta$ is more sensitive to the value selected for $T$ in the sense that small changes may result in very different values for the measured AUROCs (cf. Appendix C.4.1). In contrast, the best results are obtained for the same epsilon values of $D_\alpha$ and $D_\beta$ across all the datasets.

**Comparison in TBB**. We compare DOCTOR with MHLNB (without input pre-processing and with the softmax output in place of the logits) and SR. It is worth to emphasize that $D_\alpha$ does not coincide (in general) with SR since the former consists in the sum of squared values of all probabilities involved in the softmax. To complete the comparison, we include the results for both methods in Table 1.

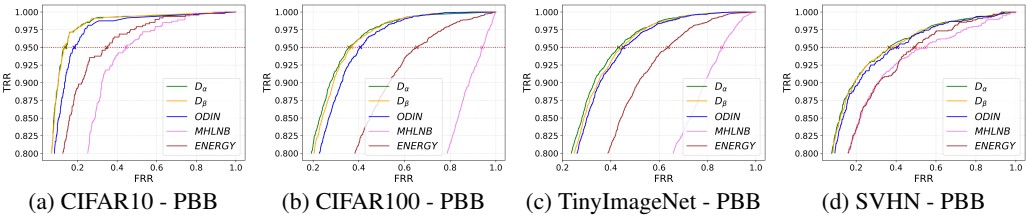

(a) CIFAR10 - PBB     (b) CIFAR100 - PBB     (c) TinyImageNet - PBB     (d) SVHN - PBB

Figure 2: ROC curves. Comparison between $D_\alpha$ ($T_\alpha = 1$ and $\epsilon_\alpha = 0.00035$), $D_\beta$ ($T_\beta = 1.5$ and $\epsilon_\alpha = 0.00035$), ODIN ($T_{\text{ODIN}} = 1.3$ and $\epsilon_{\text{ODIN}} = 0$), MHLNB ($T_{\text{MHLNB}} = 1$ and $\epsilon_{\text{MHLNB}} = 0.0002$) and ENERGY ($T_{\text{ENERGY}} = 1$ and $\epsilon_{\text{ENERGY}} = 0$). Red dashed lines mark the $95\%$ threshold of TRR.

**Comparison in PBB.** We compare DOCTOR with ODIN, MHLNB and ENERGY. We keep the same parameter setting for all the methods. In the case of DOCTOR and ODIN where temperature scaling is allowed, we test, for each dataset, $24$ different values of $\epsilon$ for each of the $11$ different values of $T$, see (Appendix C.4.2) for the set of ranges. In the case of MHLNB, which directly uses the logits, we keep $T = 1$ and we vary $\epsilon$ for each dataset. In the case of ENERGY, where no perturbation is allowed, we keep $\epsilon = 0$ and we maintain $T = 1$ (as in [24]). According to our framework, no validation samples are available; consequently, in order to be consistent across the datasets, we only report the experimental settings and values for which, on average, we obtain favorable results for all the considered domains (cf. Figure 2). In order to be fair, we update ODIN's parameters from those in [23] to new values which are more suitable to the task at hand (cf. plots in Appendix C.4.2).

DOCTOR's performance compared to ODIN's, MHLNB's and ENERGY's, are collected in Table 1 and in Figure 2. The results in the table show that noise further improves the performance of DOCTOR (cf. PBB) up to $1\%$ over our previous experiments without noise (cf. TBB) in terms of AUROC. The improvement is even more significant in terms of FRR at $95\%$ TRR: *a $4\%$ decrease is obtained in terms of predictions incorrectly rejected for* DOCTOR *when passing from TBB to PBB*. Note that only the softmax output is available when we consider the pre-trained models for AmazonFashion, AmazonSoftware and IMDb datasets; therefore, we cannot access any internal layer and test DOCTOR for values of $T$ which differ from the default value $T = 1$. Consequently, temperature scaling and input pre-processing cannot be applied in these cases and thus these datasets cannot be tested in PBB. Moreover, even in TBB, these datasets cannot be tested through MHLNB and ENERGY since the dataset on which the network was trained is not available. We provide simulations on how the range of interval for the different thresholds can affect the results in Appendix C.3.

**Misclassification detection in presence of OOD samples**. We evaluate DOCTOR's performance in misclassifcation detection considering a mixture of both in (DATASET-IN) and out-of-distribution (OOD) samples (DATASET-OUT), i.e. input samples for which the decision should not be trusted. The results are compared with ODIN. We test the two methods when one sample to reject out of five (♣), three (♢) or two (♠) is OOD. The details of the simulations, the considered dataset, and the complete experimental results are relegated Appendix C.4.3. In Table 2 we report an extract of the results for the PBB scenario in terms of *mean / standard deviation*: DOCTOR achieves, and most of the time outperforms ODIN's performance. We emphasize that, even though DOCTOR is not tuned for the OOD detection problem, it represents the best choice for deciding whether to accept or reject the prediction of the classifier also on mixed data scenarios where the percentage of OOD samples, as long as it is not dominant, can sensitively vary.

Table 1: For all methods, in TBB, we set $T = 1$ and $\epsilon = 0$; in PBB we set : $\epsilon_\alpha = \epsilon_\beta = 0.00035$, $T_\alpha = 1$, $T_\beta = 1.5$, $\epsilon_{\text{ODIN}} = 0$ and $T_{\text{ODIN}} = 1.3$, $\epsilon_{\text{MHLNB}} = 0.0002$ and $T_{\text{MHLNB}} = 1$, $\epsilon_{\text{ENERGY}} = 0$ and $T_{\text{ENERGY}} = 1$. In TBB, ODIN and SR coincide ($T = 1$ and $\epsilon = 0$).

| DATASET | METHOD | AUROC % | | FRR % (95 % TRR) | |
|---|---|---|---|---|---|
| | | TBB | PBB | TBB | PBB |
| CIFAR10 Acc. 95% | $D_\alpha$ | **94** | **95.2** | **17.9** | 13.9 |
| | $D_\beta$ | 68.5 | 94.8 | 18.6 | **13.4** |
| | ODIN | 93.8 | 94.2 | 18.2 | 18.4 |
| | SR | 93.8 | - | 18.2 | - |
| | MHLNB | 92.2 | 84.4 | 30.8 | 44.6 |
| | ENERGY | - | 91.1 | - | 34.7 |
| CIFAR100 Acc. 78% | $D_\alpha$ | **87** | **88.2** | 40.6 | **35.7** |
| | $D_\beta$ | 84.2 | 87.4 | 40.6 | 36.7 |
| | ODIN | 86.9 | 87.1 | 40.5 | 40.7 |
| | SR | 86.9 | - | **40.5** | - |
| | MHLNB | 82.6 | 50 | 66.7 | 94 |
| | ENERGY | - | 78.7 | - | 65.4 |
| TINY IMAGENET Acc. 63% | $D_\alpha$ | **84.9** | **86.1** | **45.8** | **43.3** |
| | $D_\beta$ | **84.9** | 85.3 | **45.8** | 45.1 |
| | ODIN | 84.9 | 84.9 | 45.8 | 45.3 |
| | SR | **84.9** | - | **45.8** | - |
| | MHLNB | 78.4 | 59 | 82.3 | 86 |
| | ENERGY | - | 78.2 | - | 63.7 |

| DATASET | METHOD | AUROC % | | FRR % (95 % TRR) | |
|---|---|---|---|---|---|
| | | TBB | PBB | TBB | PBB |
| SVHN Acc. 96% | $D_\alpha$ | **92.3** | **93** | **38.6** | **36.6** |
| | $D_\beta$ | 92.2 | 92.8 | 39.7 | 38.4 |
| | ODIN | 92.3 | 92.3 | 38.6 | 40.7 |
| | SR | **92.3** | - | **38.6** | - |
| | MHLNB | 87.3 | 88 | 85.8 | 54.7 |
| | ENERGY | - | 88.9 | - | 49.4 |
| AMAZON FASHION Acc. 85% | $D_\alpha$ | **89.7** | - | 27.1 | - |
| | $D_\beta$ | **89.7** | - | **26.3** | - |
| | SR | 87.4 | - | 50.1 | - |
| AMAZON SOFTWARE Acc. 73% | $D_\alpha$ | **68.8** | - | **73.2** | - |
| | $D_\beta$ | **68.8** | - | **73.2** | - |
| | SR | 67.3 | - | 86.6 | - |
| IMDB Acc. 90% | $D_\alpha$ | **84.4** | - | **54.2** | - |
| | $D_\beta$ | **84.4** | - | 54.4 | - |
| | SR | 83.7 | - | 61.7 | - |

Table 2: Same parameter setting as in table 1 (PBB) for $D_\alpha$, $D_\beta$, ODIN, ENERGY; as in [23] for ODIN$_{\text{OOD}}$ and as in [22] for MHLNB$_{\text{WB}}$. Results presented in terms of *mean / standard deviation*.

| DATASET-IN | DATASET-OUT | AUROC % | | | | | | FRR % (95 % TRR) | | | | | |
|---|---|---|---|---|---|---|---|---|---|---|---|---|---|
| | | $D_\alpha$ | $D_\beta$ | ODIN | ODIN$_{\text{OOD}}$ | ENERGY | MHLNB$_{\text{WB}}$ | $D_\alpha$ | $D_\beta$ | ODIN | ODIN$_{\text{OOD}}$ | ENERGY | MHLNB$_{\text{WB}}$ |
| CIFAR10 ♣ | iSUN | **95.4** / 0.1 | 95.1 / 0.1 | 94.6 / 0.1 | 89.6 / 0 | 92.4 / 0.1 | 54.5 / 0.1 | 14 / 0.5 | **13.5** / 0.4 | 17.2 / 0.3 | 38.9 / 0 | 32.2 / 0.1 | 92 / 0.1 |
| | TINY (RES) | **95.2** / 0.1 | 94.9 / 0.1 | 94.6 / 0.1 | 89.6 / 0 | 92.3 / 0.1 | 56.2 / 0 | **14** / 0.4 | **14** / 0.5 | 17.8 / 0.4 | 38.9 / 0 | 32.2 / 0.1 | 90.3 / 0.2 |
| CIFAR10 ◇ | iSUN | **95.5** / 0.1 | 95.3 / 0.1 | 94.9 / 0.1 | 91.5 / 0 | 92.9 / 0 | 54.5 / 0.1 | 14.4 / 0.6 | **13.4** / 0.2 | 16.8 / 0.5 | 34 / 0.1 | 27 / 1 | 92 / 0.2 |
| | TINY (RES) | **95.4** / 0.1 | 95 / 0.1 | 94.8 / 0.1 | 91.4 / 0 | 92.8 / 0 | 56.2 / 0.1 | 15 / 0.1 | **14.8** / 0.7 | 17 / 0.5 | 34.5 / 0.9 | 28.8 / 1.9 | 90 / 0.3 |
| CIFAR10 ♠ | iSUN | **95.6** / 0.1 | **95.6** / 0 | 95.4 / 0 | 93.5 / 0 | 93.6 / 0.1 | 54.6 / 0 | 15.1 / 0.1 | **13.6** / 0.5 | 16.1 / 0.2 | 30.6 / 0.4 | 25.1 / 0.2 | 92 / 0.2 |
| | TINY (RES) | **95.5** / 0.1 | 95.2 / 0.1 | 95.1 / 0.1 | 93.2 | 93.5 / 0 | 56.2 / 0.2 | **14.7** / 0.3 | 14.8 / 0.5 | 17.1 / 0.4 | 31 / 0 | 25.6 / 0.3 | 90.2 / 0.1 |

## 5   Summary and Concluding Remarks

We introduced a simple and effective method to detect misclassification errors, i.e., whether a prediction of a classifier should or should not be trusted. We provided theoretical results on the optimal statistical model for misclassification detection and we presented our empirical discriminator DOCTOR. Experiments on real (textual and visual) datasets–including OOD samples and comparison to state-of-the-art methods– demonstrate the effectiveness of our proposed methods. Whilst methods for ODD frameworks do not necessarily perform well in predicting misclassification errors, our result advances the state-of-the-art, and the main takeaway is that DOCTOR can be applied to both partially black-box (PBB) setups and totally black-box (TBB) ones. In the latter, information about the model's architecture may be undisclosed for security reason when dealing with sensitive data). DOCTOR uses all the information in the softmax output, which results in equal or better performance with respect to the other methods: the results in PBB, where we observe a reduction up to $4\%$ in terms of predictions incorrectly rejected with respect to the ones in TBB are particularly promising. Moreover, DOCTOR does not require training data and, thanks to its flexibility, it can be easily deployed in real-world scenarios. Currently, DOCTOR does not exploit information across the layers yet. Only the soft-predictions are used. Besides, the most important obstacle is the calibration of the threshold ($\gamma$) between the desired fault rejection and acceptance rates, which would require additional validation samples. However, quite often, the cost of collecting data for this operation can be prohibitive, making it difficult or too expensive to perform such calibration. As future work, we shall combine DOCTOR with other related lines of research such as: the extension to white-box incorporating additional information across the different latent codes of the model. Moreover, we shall investigate the possibility of combining the two proposed discriminators.

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
