# Supplementary Material

## A  Proofs

The following section shows the proofs for Proposition (3.1), Proposition (3.2) and Inequalities (3).

### A.1  Proof of Proposition 3.1

We recall the definition of the total variation distance when applied to distributions $P$, $Q$ on a set $\mathcal{X} \subseteq \mathbb{R}^d$ and the Scheffé's identity, Lemma 2.1 in [30]:

$$\|P - Q\|_{\text{TV}} \triangleq \sup_{\mathcal{A} \in \mathcal{B}^d} |P(\mathcal{A}) - Q(\mathcal{A})| = \frac{1}{2} \int |p_X(\mathbf{x}) - q_X(\mathbf{x})| d\mu(\mathbf{x}) \tag{19}$$

with respect to a base measure $\mu$, where $\mathcal{B}^d$ denotes the class of all Borel sets on $\mathbb{R}^d$.

*Proof.* First of all, we prove the equality for $\gamma = 1$. Let us denote with $\mathcal{A}^\star \equiv \mathcal{A}(1)$ and $\mathcal{A}^{\star c} \equiv \mathcal{A}^c(1)$ the optimal decision regions from (9). Let $\epsilon_0(\mathcal{A}^\star)$ and $\epsilon_1(\mathcal{A}^{\star c})$ be the Type-I and Type-II errors, respectively. Then,

$$\begin{aligned}
\epsilon_0(\mathcal{A}^\star) + \epsilon_1(\mathcal{A}^{\star c}) &= \int_{\mathcal{A}^\star} p_{X|E}(\mathbf{x}|0) d\mathbf{x} + \int_{\mathcal{A}^{\star c}} p_{X|E}(\mathbf{x}|1) d\mathbf{x} \\
&= \int_{\mathcal{A}^\star} \min\left\{ p_{X|E}(\mathbf{x}|0), \, p_{X|E}(\mathbf{x}|1) \right\} d\mathbf{x} \\
&\quad + \int_{\mathcal{A}^{\star c}} \min\left\{ p_{X|E}(\mathbf{x}|0), \, p_{X|E}(\mathbf{x}|1) \right\} d\mathbf{x} \\
&= \int_{\mathcal{X}} \min\left\{ p_{X|E}(\mathbf{x}|0), \, p_{X|E}(\mathbf{x}|1) \right\} d\mathbf{x} \\
&= 1 - \left\| p_{X|E=1} - p_{X|E=0} \right\|_{\text{TV}},
\end{aligned} \tag{20}$$

where the last identity follows by applying Scheffé's identity (19). From the last identity in (20) and any decision region $\mathcal{A} \subseteq \mathcal{X}$, we have

$$\begin{aligned}
1 - \left\| p_{X|E=1} - p_{X|E=0} \right\|_{\text{TV}} &= \int_{\mathcal{X}} \min\left\{ p_{X|E}(\mathbf{x}|0), \, p_{X|E}(\mathbf{x}|1) \right\} d\mathbf{x} \\
&= \int_{\mathcal{A}} \min\left\{ p_{X|E}(\mathbf{x}|0), \, p_{X|E}(\mathbf{x}|1) \right\} d\mathbf{x} \\
&\quad + \int_{\mathcal{A}^c} \min\left\{ p_{X|E}(\mathbf{x}|0), \, p_{X|E}(\mathbf{x}|1) \right\} d\mathbf{x} \\
&\leq \int_{\mathcal{A}} p_{X|E}(\mathbf{x}|0) d\mathbf{x} + \int_{\mathcal{A}^c} p_{X|E}(\mathbf{x}|1) d\mathbf{x} \\
&= \epsilon_0(\mathcal{A}) + \epsilon_1(\mathcal{A}^c).
\end{aligned} \tag{21}$$

It remains to show the last statement related to the Bayesian error of the test. Assume that $P_E(1) = P_E(0) = 1/2$. By using the last identity in (20), we have

$$\begin{aligned}
\frac{1}{2}\left[ 1 - \left\| p_{X|E=1} - p_{X|E=0} \right\|_{\text{TV}} \right] &= \frac{1}{2} \int_{\mathcal{X}} \min\left\{ p_{X|E}(\mathbf{x}|0), p_{X|E}(\mathbf{x}|1) \right\} d\mathbf{x} \\
&= \int_{\mathcal{X}} \min\left\{ p_{XE}(\mathbf{x}, E = 0), p_{XE}(\mathbf{x}, E = 1) \right\} d\mathbf{x} \\
&= \mathbb{E}_X\left[ \min\left\{ P_{E|X}(0|\mathbf{X}), P_{E|X}(1|\mathbf{X}) \right\} \right] \\
&= \frac{1}{2}\left[ \epsilon_0(\mathcal{A}^\star) + \epsilon_1(\mathcal{A}^{\star c}) \right] \\
&\equiv \inf_D \Pr\left\{ D(\mathbf{X}) \neq E \right\},
\end{aligned} \tag{22}$$

where the last identity follow by the definition of the decision regions in (9).  □

## A.2 Proof of Proposition 3.2

*Proof.* We begin by showing that

$$
\begin{aligned}
|\widehat{\mathrm{Pe}}(\mathbf{x}) - \mathrm{Pe}(\mathbf{x})| &= \left| \mathbb{E}\big[\mathbb{1}[\widehat{Y} \neq f_{\mathcal{D}_n}(\mathbf{x})]\big|\mathbf{x}\big] - \mathbb{E}\big[\mathbb{1}[Y \neq f_{\mathcal{D}_n}(\mathbf{x})]\big|\mathbf{x}\big] \right| \\
&= \left| \sum_{\{y \in \mathcal{Y} \mid y \neq f_{\mathcal{D}_n}(\mathbf{x})\}} \big[ P_{\widehat{Y}|X}(y|\mathbf{x}) - P_{Y|X}(y|\mathbf{x}) \big] \right| \\
&\leq \sum_{\{y \in \mathcal{Y} \mid y \neq f_{\mathcal{D}_n}(\mathbf{x})\}} \left| P_{\widehat{Y}|X}(y|\mathbf{x}) - P_{Y|X}(y|\mathbf{x}) \right| \\
&\leq \sum_{y \in \mathcal{Y}} \left| P_{\widehat{Y}|X}(y|\mathbf{x}) - P_{Y|X}(y|\mathbf{x}) \right| \\
&\leq 2 \left\| P_{\widehat{Y}|X}(\cdot|\mathbf{x}) - P_{Y|X}(\cdot|\mathbf{x}) \right\|_{\mathrm{TV}} \\
&\leq 2\sqrt{2\mathrm{KL}\left(P_{Y|\mathbf{x}} \| P_{\widehat{Y}|\mathbf{x}}\right)},
\end{aligned}
\tag{23}
$$

where $\|\cdot\|_{\mathrm{TV}}$ denotes the *Total Variation distance*, $\mathrm{KL}(\cdot\|\cdot)$ is the *Kullback–Leibler divergence* and the last step is due to *Pinsker's inequality*. On the other hand,

$$
\begin{aligned}
1 - \widehat{\mathrm{g}}(\mathbf{x}) &= 1 - \sum_{y \in \mathcal{Y}} P_{\widehat{Y}|X}^2(y|\mathbf{x}) \\
&= 1 - \mathbb{E}_{\widehat{Y}|X}\left[ P_{\widehat{Y}|X}(\widehat{Y}|\mathbf{x})|\mathbf{x} \right] \\
&\geq 1 - \mathbb{E}_{\widehat{Y}|X}\left[ \max_{y \in \mathcal{Y}} P_{\widehat{Y}|X}(y|\mathbf{x})|\mathbf{x} \right] \\
&= 1 - \max_{y \in \mathcal{Y}} P_{\widehat{Y}|X}(y|\mathbf{x}) \\
&\equiv \widehat{\mathrm{Pe}}(\mathbf{x}).
\end{aligned}
\tag{24}
$$

Similarly,

$$
\begin{aligned}
\widehat{\mathrm{g}}(\mathbf{x}) &= \sum_{y \in \mathcal{Y}} P_{\widehat{Y}|X}^2(y|\mathbf{x}) \\
&= P_{\widehat{Y}|X}^2(y^\star|\mathbf{x}) + \sum_{y \neq y^\star} P_{\widehat{Y}|X}^2(y|\mathbf{x}) \\
&\geq \max_{y \in \mathcal{Y}} P_{\widehat{Y}|X}^2(y|\mathbf{x}) \\
&\equiv \left( 1 - \widehat{\mathrm{Pe}}(\mathbf{x}) \right)^2,
\end{aligned}
\tag{25}
$$

where $y^\star = \arg\max_{y \in \mathcal{Y}} P_{\widehat{Y}|X}(y|\mathbf{x})$. By replacing expressions (24) and (25) in (23) we obtained the desired inequalities, which concludes the proof. $\square$

## A.3 Proof of Inequalities in (3)

*Proof.* The event can be decomposed as follows:

$$
\{\widehat{E}(\mathbf{x}) \neq E(\mathbf{x})|\mathbf{x}\} \equiv \{Y \neq \widehat{Y}\} \cap \left\{ \{\widehat{Y} = f_{\mathcal{D}_n}(\mathbf{x})\} \text{ or } \{Y = f_{\mathcal{D}_n}(\mathbf{x})\}|\mathbf{x} \right\}
\tag{26}
$$

for all $\mathbf{x} \in \mathcal{X}$. Thus,

$$
\{\widehat{E}(\mathbf{x}) \neq E(\mathbf{x})|\mathbf{x}\} \subseteq \{Y \neq \widehat{Y}|\mathbf{x}\},
\tag{27}
$$

$$
\{Y \neq \widehat{Y}\} \cap \{Y \neq f_{\mathcal{D}_n}(\mathbf{x})|\mathbf{x}\} \subseteq \{\widehat{E}(\mathbf{x}) \neq E(\mathbf{x})|\mathbf{x}\},
\tag{28}
$$

$$
\{Y \neq \widehat{Y}\} \cap \{\widehat{Y} \neq f_{\mathcal{D}_n}(\mathbf{x})|\mathbf{x}\} \subseteq \{\widehat{E}(\mathbf{x}) \neq E(\mathbf{x})|\mathbf{x}\},
\tag{29}
$$

which imply

$$\Pr\left(\{\widehat{E}(\mathbf{x}) \neq E(\mathbf{x})|\mathbf{x}\}\right) \leq \Pr\left(\{\widehat{Y} \neq Y\}|\mathbf{x}\right), \tag{30}$$

$$\mathrm{Pe}(\mathbf{x}) - \Pr\left(\{\widehat{Y} = Y\}|\mathbf{x}\right) \leq \Pr\left(\{\widehat{E}(\mathbf{x}) \neq E(\mathbf{x})\}|\mathbf{x}\right), \tag{31}$$

$$\widehat{\mathrm{Pe}}(\mathbf{x}) - \Pr\left(\{\widehat{Y} = Y\}|\mathbf{x}\right) \leq \Pr\left(\{\widehat{E}(\mathbf{x}) \neq E(\mathbf{x})\}|\mathbf{x}\right), \tag{32}$$

for all $\mathbf{x} \in \mathcal{X}$, where the last inequalities follows by noticing that $\Pr(\mathcal{A} \cap \mathcal{B}) \geq \Pr(\mathcal{A}) - \Pr(\mathcal{B}^c)$ for arbitrary measurable sets $\mathcal{A}, \mathcal{B} \subset \mathcal{X}$. This concludes the proof of these inequalities. □

## B   Logistic Regression and Gaussian Model

Throughout this section we test DOCTOR in a controlled setting were all the involved distributions are known. We refer to that setting as *logistic regression and Gaussian model* since we collect data points from Gaussians distributions and we test on the logistic regression setup.

### B.1   Theoretical analysis

Let $\mathcal{X} = \mathbb{R}^d$ be the feature space and $\mathcal{Y} = \{-1, 1\}$ be the label space. We focus on a binary classification task in which $\mathbf{X} \sim \mathcal{N}(y\boldsymbol{\mu}, \sigma^2 I)$ and $Y \sim \mathcal{U}(\mathcal{Y})$, where $\boldsymbol{\mu} \in \mathbb{R}^n$ is the mean vector, $\sigma^2 > 0$ is the variance and $I$ is the identity matrix and $\mathcal{U}(\mathcal{Y})$ denotes the uniform distribution over $\mathcal{Y}$. For a fixed $\boldsymbol{\theta} \in \mathbb{R}^d$, consider $f_{\boldsymbol{\theta}} : \mathcal{X} \to \mathcal{Y}$ s.t. $f_{\boldsymbol{\theta}}(\mathbf{x}) = \mathrm{sign}(\mathrm{sigmoid}(\mathbf{x}^T\boldsymbol{\theta}) - 1/2)$. For a given $\mathbf{x} \in \mathcal{X}$, we adapt to the current setting the definition of $E(\mathbf{x})$ in section 2 as follows:

$$\mathbb{1}\left[Y \neq f_{\boldsymbol{\theta}}(\mathbf{x})\right] = \mathbb{1}\left[Y \cdot \mathrm{sign}\left(\mathrm{sigmoid}\left(\mathbf{x}^T\boldsymbol{\theta}\right) - \frac{1}{2}\right) < 0\right]. \tag{33}$$

Let us denote by $\mathbb{1}\left[y \neq f_{\boldsymbol{\theta}}(\mathbf{x})\right]$ the realization of the random variable $E(\mathbf{x})$. We can compute the probability of classification error $\mathrm{Pe}(\mathbf{x})$ in (1) w.r.t. the true class posterior probabilities:

$$\begin{aligned}
\mathrm{Pe}(\mathbf{x}) = \mathbb{E}\left[\mathbb{1}\left[Y \neq f_{\boldsymbol{\theta}}(\mathbf{x})\right]|\mathbf{x}\right] &= \sum_{y \in \mathcal{Y}} \mathbb{1}\left[y \neq f_{\boldsymbol{\theta}}(\mathbf{x})\right] \cdot \frac{p_{\mathbf{X}|Y}(\mathbf{x}|y)P_Y(y)}{p_{\mathbf{X}}(\mathbf{x})} \\
&= \sum_{y \in \mathcal{Y}} \mathbb{1}\left[y \neq f_{\boldsymbol{\theta}}(\mathbf{x})\right] \cdot \frac{\frac{1}{2}\mathcal{N}(\mathbf{x}; y\mu, \sigma^2 I)}{\frac{1}{2}\sum_{y' \in \mathcal{Y}} \mathcal{N}(\mathbf{x}; y'\mu, \sigma^2 I)} \\
&= \frac{\sum_{y \in \mathcal{Y}} \mathbb{1}\left[y \neq f_{\boldsymbol{\theta}}(\mathbf{x})\right] \cdot \mathcal{N}(\mathbf{x}; y\mu, \sigma^2 I)}{\sum_{y \in \mathcal{Y}} \mathcal{N}(\mathbf{x}; y\mu, \sigma^2 I)}.
\end{aligned} \tag{34}$$

Following (10), the decision region corresponding to the most powerful discriminator for the logistic regression and the Gaussian model are given by

$$\mathcal{A}(\gamma) = \left\{\mathbf{x} \in \mathcal{X} : \frac{\sum_{y \in \mathcal{Y}} \mathbb{1}\left[y \neq f_{\boldsymbol{\theta}}(\mathbf{x})\right] \cdot \mathcal{N}(\mathbf{x}, y\boldsymbol{\mu}, \sigma^2 I)}{\sum_{y \in \mathcal{Y}} \mathbb{1}\left[y = f_{\boldsymbol{\theta}}(\mathbf{x})\right] \cdot \mathcal{N}(\mathbf{x}, y\boldsymbol{\mu}, \sigma^2 I)} > \gamma\right\}. \tag{35}$$

We are now able to state the optimal discriminator for this setting.

**Definition 3** (Optimal discriminator for the logistic regression and the Gaussian model). For any $0 < \gamma < \infty$ and $\mathbf{x} \in \mathcal{X}$, the optimal discriminator follows as:

$$D^{\star}(\mathbf{x}, \gamma) \triangleq \mathbb{1}\left[\sum_{y \in \mathcal{Y}} \mathbb{1}\left[y \neq f_{\boldsymbol{\theta}}(\mathbf{x})\right] \cdot \mathcal{N}(\mathbf{x}, y\boldsymbol{\mu}, \sigma^2 I) > \gamma \cdot \sum_{y \in \mathcal{Y}} \mathbb{1}\left[y = f_{\boldsymbol{\theta}}(\mathbf{x})\right] \cdot \mathcal{N}(\mathbf{x}, y\boldsymbol{\mu}, \sigma^2 I)\right]. \tag{36}$$

Since we cannot analytically evaluate Proposition 3.1, we proceed numerically in the next experiment.

## B.2 Experiments

In this section, we will numerically evaluate Proposition 3.1 via empirical estimates of Type-I and Type-II errors in expressions (5). Note that unlike section 4, in this case all the involved distributions are known and hence it is also possible to compute the *true posterior distribution* $P_{Y|X}$.

We adopt the same notation as in section 3 for DOCTOR, i.e., $D_\alpha$ and $D_\beta$ according to according to expressions (14). $D^\star$, as in Definition 3, denotes the optimal discriminator.

### B.2.1 Experimental setup and evaluation metrics

**Dataset.** We create a synthetic dataset that consists of 5000 data points drawn from $\mathcal{N}_0 \triangleq \mathcal{N}(\boldsymbol{\mu}_0, \sigma^2 I)$ and 5000 data points drawn from $\mathcal{N}_1 \triangleq \mathcal{N}(\boldsymbol{\mu}_1, \sigma^2 I)$, where $\boldsymbol{\mu}_0 = [-1 \ -1]$, $\boldsymbol{\mu}_1 = [1 \ 1]$. We consider two values for sigma, namely $\sigma = 2$ and $\sigma = 4$. These values produce two different distributions which will let us showcase the advantages of DOCTOR. To each data point $\mathbf{x}$ is assigned as class 0 or 1 depending on whether $\mathbf{x} \sim \mathcal{N}_0$ or $\mathbf{x} \sim \mathcal{N}_1$, respectively. The aforementioned dataset is divided into a training set, i.e. $\mathcal{D}_n = \{(\mathbf{x}_1, y_1), \dots, (\mathbf{x}_n, y_n)\}$ where $n = 6700$, and a testing set, i.e. $\mathcal{T}_m = \{(\mathbf{x}_{n+1}, y_{n+1}), \dots, (\mathbf{x}_{n+m}, y_{n+m})\}$ where $m = 3300$.

Table 3: Accuracy on the test set: $f_{\boldsymbol{\theta}_i}$ for $i = 1, \dots, 8$ represents the $i$-th model in $\mathcal{F}$, $f_{avg}$ is the arithmetic mean of the accuracy over each $f_{\boldsymbol{\theta}_i} \in \mathcal{F}$. The value $f_{avg}^\star$ represents the accuracy Bayesian classifier averaged on the test set corresponding to the 8 splits. We show results for both standard deviations, namely $\sigma = 2$ and $\sigma = 4$.

| CLASSIFIER | ACCURACY% | |
|---|---|---|
| | $\sigma = 2$ | $\sigma = 4$ |
| $f_{\boldsymbol{\theta}_1}$ | 82 | 65 |
| $f_{\boldsymbol{\theta}_2}$ | 83 | 77 |
| $f_{\boldsymbol{\theta}_3}$ | 82 | 77 |
| $f_{\boldsymbol{\theta}_4}$ | 82 | 76 |
| $f_{\boldsymbol{\theta}_5}$ | 83 | 76 |
| $f_{\boldsymbol{\theta}_6}$ | 81 | 66 |
| $f_{\boldsymbol{\theta}_7}$ | 82 | 76 |
| $f_{\boldsymbol{\theta}_8}$ | 83 | 83 |
| $f_{avg}$ | 82 | 74 |
| $f_{avg}^\star$ | 83 | 78 |

**Training configuration.** We use a linear classifier, with one hidden layer, sigmoid activation function and binary cross entropy loss. The neural network is trained with gradient descent considering learning rate $r = 0.1$. Specifically, we train our network for 5 epochs. We randomly split our dataset 8 times, each time keeping $n$ samples to train, and $m$ to test. We consider the same model architecture (described above) for each split and we come up with 8 different binary discriminators $\mathcal{F} = \{f_{\boldsymbol{\theta}_1}, \dots, f_{\boldsymbol{\theta}_8}\}$. Since in this example all the involved distributions are known, we compute the optimal predictor, i.e. the Bayes classifier, and we denote it with $f^\star$. The value $f_{avg}^\star$ reported in table 3, represents its accuracy averaged on the test set corresponding to the 8 splits.

**Accuracy of trained networks.** In table 3 the accuracy of $f^\star$ and the models in $\mathcal{F}$ on the test set.

**Evaluation metric.** We consider the same metric as in section 4.2.

### B.2.2 Numerical evaluation of Proposition 3.1

To evaluate Proposition 3.1 we proceed in a Monte Carlo fashion by computing Type-I and Type-II errors for each of the network in $\mathcal{F}$ and then averaging over the results. Schematically, consider any $f_{\boldsymbol{\theta}_i} \in \mathcal{F}$ and $\gamma = 1$, we compute:

1. $\mathcal{A}_i \triangleq \mathcal{A}_i(1)$ as defined in eq. (35) and its complement $\mathcal{A}_i^c$.

2. For each classifier $f_{\boldsymbol{\theta}_i} \in \mathcal{F}$, $\mathcal{T}_{E=1;\boldsymbol{\theta}_i} \triangleq \{(\mathbf{x}, y) \in \mathcal{T}_m \mid y \neq f_{\boldsymbol{\theta}_i}(\mathbf{x})\}$ represents the set of mis-classified test samples, and $\mathcal{T}_{E=0;\boldsymbol{\theta}_i} \triangleq \{(\mathbf{x}, y) \in \mathcal{T}_m \mid y = f_{\boldsymbol{\theta}_i}(\mathbf{x})\}$ is the set of correctly classified test samples.

3. $\mathcal{FR}_i \triangleq \{(\mathbf{x}, y) \in \mathcal{T}_{E=0;\boldsymbol{\theta}_i} : \mathbf{x} \in \mathcal{A}_i\}$, $\mathcal{TR}_i \triangleq \{(\mathbf{x}, y) \in \mathcal{T}_{E=1;\boldsymbol{\theta}_i} : \mathbf{x} \in \mathcal{A}_i\}$, $\mathcal{FA}_i \triangleq \{(\mathbf{x}, y) \in \mathcal{T}_{E=1;\boldsymbol{\theta}_i} : \mathbf{x} \in \mathcal{A}_i^c\}$ and $\mathcal{TA}_i \triangleq \{(\mathbf{x}, y) \in \mathcal{T}_{E=0;\boldsymbol{\theta}_i} : \mathbf{x} \in \mathcal{A}_i^c\}$, i.e. the set of false rejections, true rejections, false acceptances and true acceptance, respectively.

4. $\epsilon_0(\mathcal{A}_i) \triangleq \frac{|\mathcal{FR}_i|}{|\mathcal{T}_{E=0;\boldsymbol{\theta}_i}|}$ and $\epsilon_1(\mathcal{A}_i^c) \triangleq \frac{|\mathcal{FA}_i|}{|\mathcal{T}_{E=1;\boldsymbol{\theta}_i}|}$, i.e. Type-I and Type-II errors.

At the end of $|\mathcal{F}|$ iterations, we empirically estimate Type-I and Type-II errors of Proposition 3.1 as follows

$$\epsilon_0(\mathcal{A}) \approx \frac{1}{|\mathcal{F}|}\sum_{i=1}^{|\mathcal{F}|}\epsilon_0(\mathcal{A}_i) = 0.0607 \quad \text{and} \quad \epsilon_1(\mathcal{A}^c) \approx \frac{1}{|\mathcal{F}|}\sum_{i=1}^{|\mathcal{F}|}\epsilon_1(\mathcal{A}_i^c) = 0.7389.$$

### B.2.3 FRR versus TRR

We present the experimental results obtained by running experiments similar to those described in section 4 considering the experimental setup in B.2.1 in TBB. In addition to the usual discriminators, we are going to consider the optimal discriminator $D^\star$, as in Definition 3.

**DOCTOR: comparison between $D^\star$, $D_\alpha$ and $D_\beta$.** Let us present the result obtained with DOCTOR showing how $D^\star$ (36) works compared to $D_\alpha$ and $D_\beta$ in (14) when they have to decide whether to trust or not the decision made by a classifier. We test the discriminators on the dataset constructed as in B.2.1 by considering $\sigma = 2$. Let us analyze fig. 3a: we apply each discriminator to all the classifiers in $\mathcal{F}$. The obtained ROCs are represented by the colored areas. Inside each area the mean ROC is represented by the thick line. $D_\alpha$ and $D_\beta$ reach same results as the colored areas and the thick lines are overlapped. For a given $\mathbf{x} \in \mathcal{X}$, we recall that $D^\star$ uses Pe($\mathbf{x}$) (1) whilst $D_\alpha$ and $D_\beta$ uses $1 - \widehat{g}(\mathbf{x})$ (11) and $\widehat{\text{Pe}}(\mathbf{x})$ (2), respectively. $D^\star$ always outperforms both $D_\alpha$ and $D_\beta$ since it relies on the probability of classification error based on $P_{Y|X}$ while $D_\alpha$ and $D_\beta$ use $P_{\widehat{Y}|X}$.

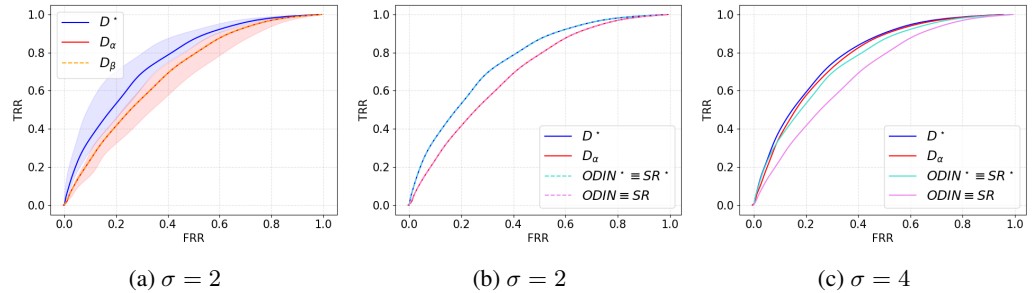

(a) $\sigma = 2$       (b) $\sigma = 2$       (c) $\sigma = 4$

Figure 3: ROC curves for $D^\star$, $D_\alpha$ and $D_\beta$, respectively. We denote by SR$^\star$ the softmax response method based on $P_{Y|X}$. Since in this case $T = 1$ and $\epsilon = 0$, SR $\equiv$ ODIN as well as SR$^\star$ $\equiv$ ODIN$^\star$. (a) We apply each discriminator to all the classifiers in $\mathcal{F}$. The obtained ROCs are represented by the colored areas. Inside each area the mean ROC is represented by the thick line. Orange and red areas completely overlap as well as the mean ROC. $D^\star$ always outperforms both $D_\alpha$ and $D_\beta$ as expected. In (b) $D^\star$ and SR$^\star$ overlap (as also $D_\alpha$ and SR), instead in (c) where $\sigma = 4$ and hence the distribution is smoother, SR discards useful information and indeed both $D^\star$ and $D_\alpha$ outperform SR.

**Comparison between $D^\star$, $D_\alpha$, ODIN and SR.** We conclude this section by investigating how our competitors, namely ODIN and SR, work in this setting.

From now on, we will put ODIN $\equiv$ SR to mean that the two methods coincide (remember we set $T = 1$ and $\epsilon = 0$ for all the simulations). We show the results of the comparison in fig. 3: fig. 3b considers data points from $\mathcal{N}(y\boldsymbol{\mu}, 2^2 I)$ whilst fig. 3c consider data points from $\mathcal{N}(y\boldsymbol{\mu}, 4^2 I)$. If in fig. 3b we cannot see an advantage in using $D_\alpha$ in place of SR, the situation is totally different in fig. 3c, where $D^\star$ and $D_\alpha$ clearly outperform the competitors. We would like to recall that DOCTOR uses all the softmax output while SR only uses the maximum value of the softmax output. Therefore, when the underlying distribution $p_{XY}$ is more smooth like in fig. 3c, SR discards useful information. As result, not only $D^\star$ outperforms SR$^\star$ but even $D_\alpha$ does the same. This is more

Table 4: AUROCs: the values for $D_\alpha$, $D_\beta$, SR, and ODIN correspond to the results for the thick lines in fig. 3. $D^\star$ and ODIN$^\star$ $\equiv$ SR$^\star$ are obtained using $P_{Y|X}$.

| | | | | **AUROC** % | |
|---|---|---|---|---|---|
| $\sigma$ | $D^\star$ | $D_\alpha$ | $D_\beta$ | SR $\equiv$ ODIN | SR$^\star$ $\equiv$ ODIN$^\star$ |
| 2 | **76** | 70 | 70 | 70 | 76 |
| 4 | **79** | 78 | 78 | 70 | 76 |

Table 5: AUROCs and FRR at 95% TRR obtained via $D_\alpha$, $D_\beta$, ODIN, SR and MHLNB for CIFAR10 considering different size for $\Gamma_{D_\alpha \text{ or } D_\beta}$, $\Delta_{\text{ODIN or SR}}$ and $Z_{\text{MHLNB}}$ in both TBB and PBB. The column INTERVAL SIZE represents the number of equidistant values considered in the sets defined in (37), (38), (39), (40) and in (41), respectively.

| INTERVAL SIZE | METHOD | TBB | | PBB | | INTERVAL SIZE | METHOD | TBB | | PBB | |
|---|---|---|---|---|---|---|---|---|---|---|---|
| | | AUROC | FRR (95% TRR) | AUROC | FRR (95% TRR) | | | AUROC | FRR (95% TRR) | AUROC | FRR (95% TRR) |
| 10 | $D_\alpha$ | 69.8 | 91.6 | 77.4 | 88.4 | 1000 | $D_\alpha$ | 91.3 | 53.1 | 94.7 | 13.8 |
| | $D_\beta$ | 50 | 69.7 | 79.8 | 86.2 | | $D_\beta$ | 66.5 | 48.3 | 94.8 | 13.4 |
| | ODIN | 75.7 | 89.3 | 81.4 | 85.4 | | ODIN | 92.5 | 28.9 | 94 | 18.3 |
| | SR | 75.7 | 89.3 | - | - | | SR | 92.5 | 28.9 | - | - |
| | MHLNB | 76.6 | 88.8 | 83.2 | 47.1 | | MHLNB | 92.2 | 35.3 | 84.4 | 44.5 |
| 100 | $D_\alpha$ | 85.1 | 80.6 | 92.5 | 42.6 | 10000 | $D_\alpha$ | 93.7 | 18.4 | 95.2 | 13.9 |
| | $D_\beta$ | 61.8 | 63.4 | 94.1 | 13.8 | | $D_\beta$ | 68.5 | 18.6 | 94.8 | 13.4 |
| | ODIN | 88 | 73.5 | 91.5 | 49.9 | | ODIN | 93.9 | 18 | 94.2 | 18.4 |
| | SR | 88 | 73.5 | - | - | | SR | 93.9 | 18 | - | - |
| | MHLNB | 88.3 | 72.6 | 84.4 | 44.6 | | MHLNB | 92.1 | 31 | 84.4 | 44.6 |

evident if we look to table 3, where for $\sigma = 4$ we notice an improvement in terms of AUROC from 70% to 78% when passing from SR to $D_\alpha$.

## C  Supplementary Results of Section 4

### C.1  Experimental environment

We run each experiment on a machine equipped with an Intel(R) Xeon(R) CPU E5-2623 v4, 2.60GHz clock frequency, and a GeForce GTX 1080 Ti GPU. The execution time for the execution the tests are the following (interval size 10000):

TBB. $D_\alpha$: 12.5 s. $D_\beta$: 13.6 s. SR: 15.9 s. MHLNB: 15.9 s.

PBB: $D_\alpha$: 13 s. $D_\beta$: 25.7 s. ODIN: 14.7 s. MHLNB: 32.22 s.

### C.2  On the input pre-processing in DOCTOR

In the following we further study DOCTOR-specific input pre-processing techniques allowed under PBB. We focus on $D_\beta$ since for $D_\alpha$ the reasoning is the same. Formally, let $\mathbf{x}_0 \in \mathcal{X}$ be a testing sample. We are looking for the minimum way to perturb the input such that the discriminator value at $\mathbf{x}_0$ is increased:

$$r^* = \min_{r \text{ s.t. } \|r\|_\infty \leq \epsilon} -\log\left(\frac{\widehat{\text{Pe}}(\mathbf{x}_0 + r)}{1 - \widehat{\text{Pe}}(\mathbf{x}_0 + r)}\right),$$

or equivalently, we are looking to the sample $\widetilde{\mathbf{x}}_0^\beta$ in the $\epsilon$-ball around $\mathbf{x}_0$ which maximize the discriminator value at $\widetilde{\mathbf{x}}_0^\beta$:

$$\widetilde{\mathbf{x}}_0^\beta = \mathbf{x}_0 - \epsilon \times \text{sign}\left[-\nabla_{\mathbf{x}_0} \log\left(\frac{\widehat{\text{Pe}}(\mathbf{x}_0)}{1 - \widehat{\text{Pe}}(\mathbf{x}_0)}\right)\right].$$

Note that, because of eq. (1)

$$-\log\left(\frac{\widehat{\text{Pe}}(\mathbf{x}_0)}{1 - \widehat{\text{Pe}}(\mathbf{x}_0)}\right) = -\log\left(\frac{1 - P_{\widehat{Y}|X}(f_{\mathcal{D}_n}(\mathbf{x}_0)|\mathbf{x}_0)}{P_{\widehat{Y}|X}(f_{\mathcal{D}_n}(\mathbf{x}_0)|\mathbf{x}_0)}\right)$$
$$= -\log(1 - P_{\widehat{Y}|X}(f_{\mathcal{D}_n}(\mathbf{x}_0)|\mathbf{x}_0)) + \log(P_{\widehat{Y}|X}(f_{\mathcal{D}_n}(\mathbf{x}_0)|\mathbf{x}_0))$$
$$= -\log(1 - P_{\widehat{Y}|X}(f_{\mathcal{D}_n}(\mathbf{x}_0)|\mathbf{x}_0)) - \log \text{SODIN}(\mathbf{x}_0).$$

### C.3  On the effect the intervals considered for $\gamma$, $\delta$ and $\zeta$ have on the AUROC computation

Let us consider the AUROC as a performance measure for the discriminators. The computation of the AUROC of $D_\alpha$, as well as those of ODIN and SR, heavily depend on the choice of the range

values for the decision region thresholds. In the following paragraph, we will discuss how we chose these ranges, namely $\gamma \in \Gamma_{D_\alpha \text{ or } D_\beta} \subseteq \mathbb{R}$, $\delta \in \Delta_{\text{ODIN or SR}} \subseteq [0, 1]$ and $\zeta \in Z_{\text{MHLNB}} \subseteq \mathbb{R}$. In the experiments of section 4, we therefore proceed by fixing the aforementioned ranges as follows:

$$\Gamma_{D_\alpha} \triangleq \left[ \min_{(\mathbf{x},y) \in \mathcal{T}_m} \frac{1 - \widehat{g}(\mathbf{x})}{\widehat{g}(\mathbf{x})}, \max_{(\mathbf{x},y) \in \mathcal{T}_m} \frac{1 - \widehat{g}(\mathbf{x})}{\widehat{g}(\mathbf{x})} \right], \tag{37}$$

$$\Gamma_{D_\beta} \triangleq \left[ \min_{(\mathbf{x},y) \in \mathcal{T}_m} \frac{\widehat{Pe}(\mathbf{x})}{1 - \widehat{Pe}(\mathbf{x})}, \max_{(\mathbf{x},y) \in \mathcal{T}_m} \frac{\widehat{Pe}(\mathbf{x})}{1 - \widehat{Pe}(\mathbf{x})} \right], \tag{38}$$

$$\Delta_{\text{ODIN}} \triangleq \left[ \min_{(\mathbf{x},y) \in \mathcal{T}_m} \text{SODIN}(\mathbf{x}), \max_{(\mathbf{x},y) \in \mathcal{T}_m} \text{SODIN}(\mathbf{x}) \right], \tag{39}$$

$$\Delta_{\text{SR}} \triangleq \left[ \min_{(\mathbf{x},y) \in \mathcal{T}_m} \text{SR}(\mathbf{x}), \max_{(\mathbf{x},y) \in \mathcal{T}_m} \text{SR}(\mathbf{x}) \right], \tag{40}$$

$$Z_{\text{MHLNB}} \triangleq \left[ \min_{(\mathbf{x},y) \in \mathcal{T}_m} \text{M}(\mathbf{x}), \max_{(\mathbf{x},y) \in \mathcal{T}_m} \text{M}(\mathbf{x}) \right]. \tag{41}$$

Secondly, we fix the number of values to consider in $\Gamma_{D_\alpha \text{ or } D_\beta}$, $\Delta_{\text{ODIN or SR}}$ and $Z_{\text{MHLNB}}$: we test the AUROCs for CIFAR10 for different values of the size of $\Gamma_{D_\alpha \text{ or } D_\beta}$, $\Delta_{\text{ODIN or SR}}$ and $Z_{\text{MHLNB}}$ in both TBB and PBB scenarios. The results are collected in table 5. Let us denote by $I$ a generic interval between the ones of eq. (37), eq. (38), eq. (39), eq. (40) and eq. (41), throughout the experiments we set the size of $I$ to $(\max I - \min I) * 10000$.

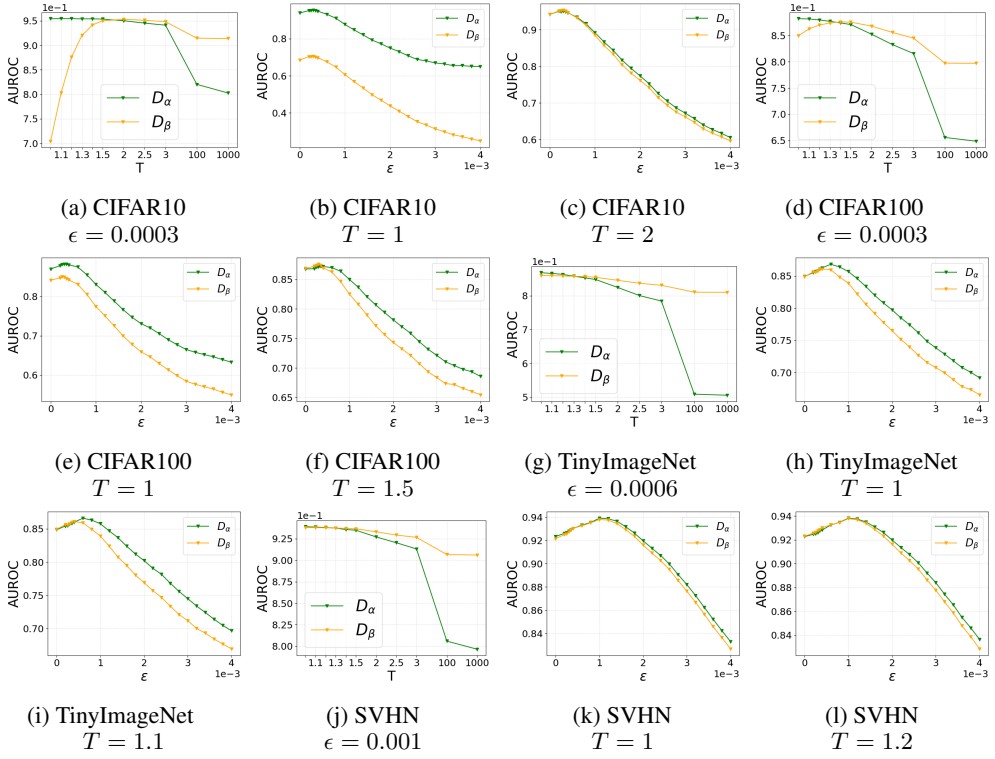

Figure 4: Comparison of AUROCs obtained via $D_\alpha$ (in green) and via $D_\beta$ (in orange) for different values of $T$ and $\epsilon$.

## C.4  Additional plots and results

In the next sections, we show graphically the set of results obtained from the experiments in section 4.3. We first specify the range of values for the parameters $T$ and $\epsilon$ considered throughout the experiments. For temperature scaling, $T$ is selected among

$\{1, 1.1, 1.2, 1.3, 1.4, 1.5, 2, 2.5, 3, 100, 1000\}$, whilst for input pre-processing, $\epsilon$ is selected among $\{0, .0002, .00025, .0003, .00035, .0004, .0006, .0008, .001, .0012, .0014, .0016, .0018, .002, .0022, .0024, .0026, .0028, .003, .0032, .0034, .0036, .0038, .004\}$.

### C.4.1 Comparison $D_\alpha$ and $D_\beta$

We include the plots for DOCTOR: *comparison between $D_\alpha$ and $D_\beta$* (section 4.3). In fig. 4a, fig. 4d, fig. 4g and fig. 4j, we set $\epsilon$ at its best value which is found to coincide in the case of $D_\alpha$ and $D_\beta$. In fig. 4b, fig. 4e, fig. 4h and fig. 4k we do the opposite and we set $T$ to its best value w.r.t. $D_\alpha$ whilst in fig. 4c, fig. 4f, fig. 4i and fig. 4l, the value of $T$ is chosen w.r.t. the best value for $D_\beta$.

### C.4.2 Comparison $D_\alpha$, $D_\beta$, ODIN and MHLNB

We conclude by showing in fig. 5 the test results obtained by varying $T$ and $\epsilon$ in PBB for all the methods. We present 4 groups of plots (one for each image dataset) and in each plot we pick $T$ from $\{1, 1.3, 1.5, 1000\}$ (the values selected for $D_\alpha$, $D_\beta$, ODIN and MHLNB table 1) and we let $\epsilon$ vary.

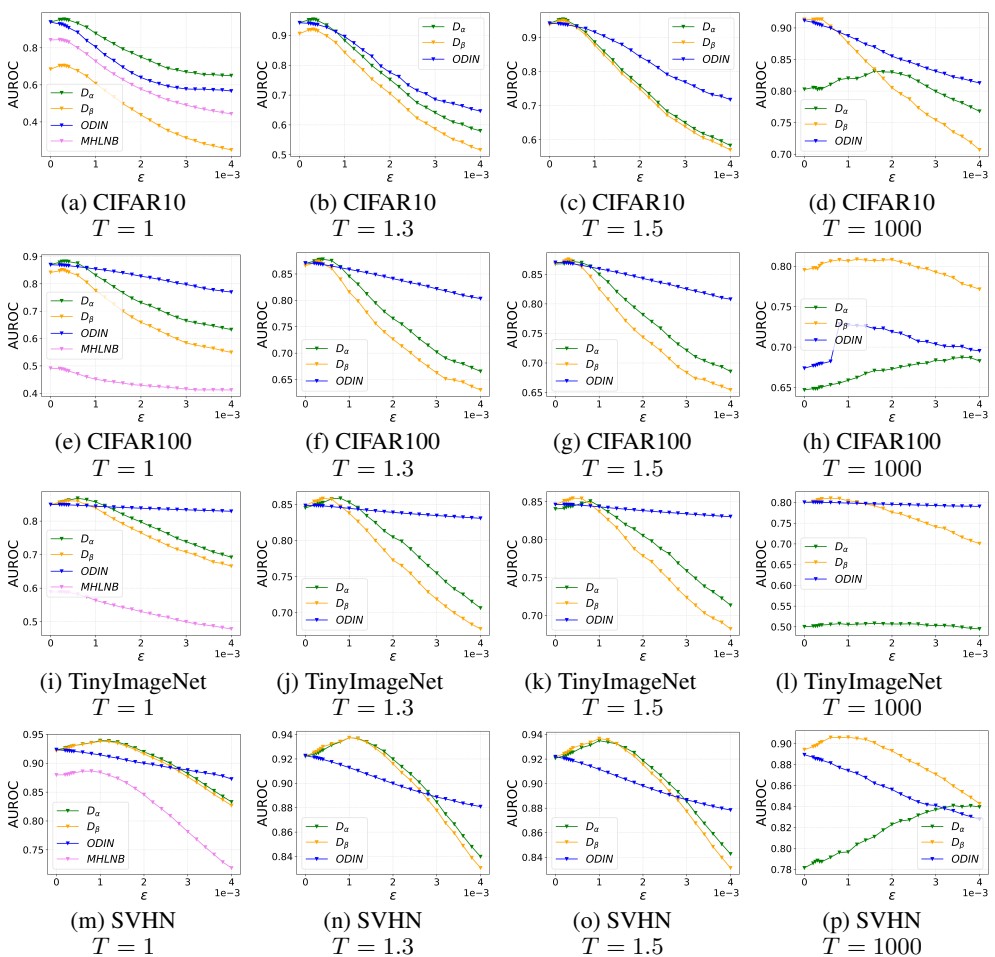

Figure 5: Overall results PBB overall datasets (by varying $T$ and $\epsilon$)

### C.4.3 Misclassification detection in presence of out-of-distribution samples

We include in table 6 the results of all the simulations carried out for detecting misclassification detection in presence of out-of-distribution samples. The experimental setting is reported in section 4.2.

## C.5 DOCTOR for pure OOD detection

It is worth emphasizing that DOCTOR is not targeting OOD detection, which is a rather different problem from the one investigated in this paper. So we did not optimize an ad-hoc input perturbation for DOCTOR within the OOD detection setup, i.e. we kept the same input perturbation proposed for the misclassification detection task. The baseline results reported in table 7 show that DOCTOR is competitive for OOD detection as well since it can reach similar scores or even outperform the baseline (e.g., the simulations with LSUN (CROP) show an improvement of the results of $3.3\%$ in terms of FRR $\%$). We indicate the methods together with their parameter setting. ODIN$_{OOD}$ denotes the same parameter setting as in [23].

### C.5.1 DOCTOR in presence of OOD samples that are similar to in-distribution ones

We tested DOCTOR in pure OOD setting, considering CIFAR100 as in-distribution and CIFAR10 as out-distribution. The results below show that DOCTOR optimized as in the following paper outperforms ODIN (optimized as described in [23]) and ENERGY. This is particularly promising as it shows that DOCTOR, without performing any training and without been particularly optimized for OOD detection, can perform well on a wider variety of problems.

### C.6 Some observations on the white-box scenario (WB)

It is worth clarifying the results in Table 9 to motivate the performance obtained using the Mahalanobis-based discriminator (MHLNB - WB) for the misclassification detection problem and the issues it raises. First of all, we emphasize that given a network and an input sample DOCTOR only needs to access the logits output of the network in order to perform the detection. On the contrary, the detector based on Mahalanobis distance consists of 3 steps:

- Estimation of the class mean and covariance matrix;
- Features extraction according to the Mahalanobis score function;
- Aggregation of the scores obtained layer by layer in order to obtain a decision a rule for the discriminator.

Clearly, the Mahalanobis distance-based method requires additional samples compared to DOCTOR. Although estimating the mean and the covariance matrix is possible by exploiting samples from the benchmark training set (e.g. CIFAR10, CIFAR100, ...), this method still needs additional (different from training) samples for learning the linear regressor intended to distinguish between correctly (positive) and incorrectly (negative) classified samples. In order to generate the negative samples, we consider the use of adversarial examples generated through Projected Gradient Descent Attack (magnitude of the perturbation 0.0031), which does not assume any knowledge about the test set.

Table 6: In PBB we set $\epsilon_\alpha = 0.00035$ and $T_\alpha = 1$, $\epsilon_\beta = 0.00035$ and $T_\beta = 1.5$, $\epsilon_{\text{ODIN}} = 0$ and $T_{\text{ODIN}} = 1.3$. By ODIN$_{\text{ood}}$, we mean ODIN with the parameter setting as in [23]. Since we proceed in a Monte Carlo fashion, the results are reported in terms of *mean / standard deviation*. In TBB for by ODIN we report the results of SR, since both methods coincide when $T = 1$ and $\epsilon = 0$.

| DATASET (IN) | DATASET (OUT) | SCENARIO | AUROC % | | | | FRR % (95 % TRR) | | | |
|---|---|---|---|---|---|---|---|---|---|---|
| | | | $D_\alpha$ | $D_\beta$ | ODIN | ODIN$_{\text{ood}}$ | $D_\alpha$ | $D_\beta$ | ODIN | ODIN$_{\text{ood}}$ |
| CIFAR10 ♣ | ISUN | PBB | **95.4** / 0.1 | 95.1 / 0.1 | 94.6 / 0.1 | 89.6 / 0 | 14 / 0.5 | **13.5** / 0.4 | 17.2 / 0.3 | 38.9 / 0 |
| | | TBB | **94.6** / 0 | 69.3 / 0.1 | 94.5 / 0.1 | - | 17.7 / 0.1 | **17.7** / 0.1 | 17.7 / 0 | - |
| | LSUN (CROP) | PBB | **95.5** / 0.1 | 95.1 / 0 | 94.7 / 0 | 92.6 / 0 | 13.1 / 0.5 | **13** / 0.2 | 17.3 / 0 | 31.9 / 0.1 |
| | | TBB | **94.4** / 0.1 | 69.2 / 0.1 | **94.4** / 0 | - | 17.6 / 0.2 | 17.6 / 0.2 | 17.7 / 0.2 | - |
| | LSUN (RESIZE) | PBB | **95.4** / 0.1 | 95.1 / 0 | 94.8 / 0 | 89.6 / 0 | 13.4 / 0.6 | **13.2** / 0.3 | 17 / 0.3 | 38.9 / 0 |
| | | TBB | **94.6** / 0.1 | 69.3 / 0.1 | 94.5 / 0.1 | - | **17.8** / 0.1 | **17.8** / 0.1 | **17.8** / 0.1 | - |
| | TINY (CROP) | PBB | **95.4** / 0 | 95.1 / 0.1 | 94.7 / 0 | 89.6 / 0 | 13.4 / 0.4 | **13** / 0.2 | 17.2 / 0.3 | 38.9 / 0 |
| | | TBB | **94.6** / 0 | 69.4 / 0.1 | **94.6** / 0 | - | 17.8 / 0.1 | 17.8 / 0.1 | 17.8 / 0.1 | - |
| | TINY (RES) | PBB | **95.2** / 0.1 | 94.9 / 0 | 94.6 / 0.1 | 89.6 / 0 | **14** / 0.4 | **14** / 0.5 | 17.8 / 0.4 | 38.9 / 0 |
| | | TBB | **94.4** / 0.1 | 69.2 / 0 | **94.4** / 0 | - | 17.8 / 0.1 | 17.8 / 0.1 | 17.8 / 0.1 | - |
| CIFAR100 ♣ | ISUN | PBB | **86.5** / 0.2 | 85.8 / 0 | 85.6 / 0.2 | 79 / 0.1 | **45.3** / 1 | 46.1 / 0.5 | 46.8 / 1 | 65.9 / 0.4 |
| | | TBB | **85.6** / 0.1 | 82.7 / 0.1 | 85.5 / 0.1 | - | 46.9 / 0.4 | **46.8** / 0.4 | **46.8** / 0.4 | - |
| | LSUN (CROP) | PBB | **89.1** / 0 | 88.5 / 0.1 | 88 / 0.1 | 80.6 / 0 | **35.6** / 0.4 | 35.7 / 0.2 | 39.9 / 0.3 | 65.1 / 0 |
| | | TBB | **87.9** / 0.1 | 84.9 / 0.1 | 87.7 / 0.1 | - | **39.8** / 0.6 | **39.8** / 0.6 | **39.8** / 0.6 | - |
| | LSUN (RESIZE) | PBB | **86.8** / 0.1 | 86.2 / 0.1 | 86 / 0.1 | 79.1 / 0.1 | 44.4 / 0.9 | **44.4** / 0.6 | 45.3 / 0.3 | 65.4 / 0.3 |
| | | TBB | **85.8** / 0.1 | 82.9 / 0.1 | 85.7 / 0.1 | - | 45.9 / 0.5 | **45.8** / 0.5 | **45.8** / 0.5 | - |
| | TINY (CROP) | PBB | **88.4** / 0.1 | 87.8 / 0.1 | 87.6 / 0.1 | 81.8 / 0.1 | 38.2 / 0.4 | **37.8** / 0.9 | 40.6 / 0.5 | 63.4 / 0.1 |
| | | TBB | **87.2** / 0.1 | 84.2 / 0.1 | 87 / 0.1 | - | **42** / 0.6 | **42** / 0.6 | **42** / 0.6 | - |
| | TINY (RES) | PBB | **86.8** / 0.1 | 86.3 / 0.1 | 85.9 / 0.1 | 79.2 / 0.1 | 44 / 0.1 | **43.6** / 0.2 | 45.9 / 1.2 | 65.8 / 0.3 |
| | | TBB | **85.9** / 0.2 | 83 / 0.2 | 85.8 / 0.2 | 85.8 / 0.2 | **45.7** / 1.3 | **45.7** / 1.3 | **45.7** / 1.3 | - |
| CIFAR10 ◇ | ISUN | PBB | **95.5** / 0.1 | 95.3 / 0.1 | 94.9 / 0.1 | 91.5 / 0 | 14.4 / 0.6 | **13.4** / 0.2 | 16.8 / 0.5 | 34 / 0.1 |
| | | TBB | **95** / 0 | 69.6 / 0 | 94.9 / 0.1 | - | **16.4** / 0.2 | **16.4** / 0.2 | **16.4** / 0.2 | - |
| | LSUN (CROP) | PBB | **95.8** / 0.1 | 95.5 / 0.1 | 95 / 0.1 | 93.9 / 0.1 | **12.4** / 0.2 | 12.6 / 0.1 | 16.1 / 0.4 | 24.8 / 0.1 |
| | | TBB | **94.8** / 0.1 | 69.6 / 0.1 | **94.8** / 0.1 | - | 16.7 / 0.4 | 16.8 / 0.4 | **16.6** / 0.4 | - |
| | LSUN (RESIZE) | PBB | **95.8** / 0 | 95.6 / 0 | 95.2 / 0 | 91.6 / 0 | **12.9** / 0.5 | **12.9** / 0.3 | 15.8 / 0.2 | 33.9 / 0 |
| | | TBB | **95** / 0 | 69.7 / 0.1 | **95** / 0.1 | - | **16.4** / 0.2 | **16.4** / 0.3 | **16.4** / 0.2 | - |
| | TINY (CROP) | PBB | **95.8** / 0.1 | 95.5 / 0.1 | 95.2 / 0.1 | 91.5 / 0 | **12.8** / 0.7 | 12.9 / 0.5 | 16 / 0 | 33.9 / 0 |
| | | TBB | **95** / 0.2 | 69.8 / 0.1 | **95** / 0.1 | - | **16.4** / 0.2 | 16.5 / 0.2 | **16.4** / 0.2 | - |
| | TINY (RES) | PBB | **95.4** / 0.1 | 95 / 0.1 | 94.8 / 0.1 | 91.4 / 0 | 15 / 0.1 | **14.8** / 0.7 | 17 / 0.5 | 34.5 / 0.9 |
| | | TBB | **94.6** / 0.2 | 69.3 / 0.2 | **94.6** / 0.2 | - | 18.1 / 1 | 18.1 / 1.1 | **18** / 1 | - |
| CIFAR100 ◇ | ISUN | PBB | **84.8** / 0.1 | 84.4 / 0.2 | 84.6 / 0.1 | 80.8 / 0.2 | 53.6 / 1 | **51.2** / 0.2 | 51.3 / 0.1 | 63.5 / 0.3 |
| | | TBB | **84.1** / 0.1 | 81.2 / 0.1 | 84 / 0.1 | - | **52.5** / 0.5 | **52.5** / 0.5 | **52.5** / 0.5 | - |
| | LSUN (CROP) | PBB | **89.9** / 0.1 | 89.6 / 0 | 89 / 0 | 84.1 / 0 | **35.2** / 0.7 | 35.4 / 0.2 | 39.3 / 0.1 | 62.2 / 0 |
| | | TBB | **88.7** / 0.1 | 85.7 / 0 | 88.5 / 0.1 | - | **38.8** / 0.5 | **38.8** / 0.5 | **38.8** / 0.4 | - |
| | LSUN (RESIZE) | PBB | **85.3** / 0.3 | 85.1 / 0.2 | 84.9 / 0.1 | 81.1 / 0 | 51.6 / 0.9 | **48.8** / 1 | 49.2 / 0.7 | 63.3 / 0.1 |
| | | TBB | **84.6** / 0.2 | 81.8 / 0.2 | **84.6** / 0.1 | - | **50.6** / 0.8 | 50.7 / 0.8 | **50.6** / 0.8 | - |
| | TINY (CROP) | PBB | **88.2** / 0 | 88.1 / 0.2 | 87.7 / 0.1 | 84.8 / 0.1 | 41.2 / 0.3 | **40.2** / 0.6 | 42.3 / 0.4 | 59 / 0.2 |
| | | TBB | **87.7** / 0.1 | 84.7 / 0.1 | 87.5 / 0.1 | - | **41.8** / 0.5 | **41.8** / 0.5 | **41.8** / 0.5 | - |
| | TINY (RES) | PBB | **85.4** / 0.2 | 84.8 / 0.2 | 85.1 / 0.3 | 81.2 / 0.1 | 51.8 / 1.6 | 52 / 0.8 | **50.4** / 0.9 | 63.3 / 0.2 |
| | | TBB | **84.8** / 0.1 | 81.9 / 0.1 | 84.7 / 0.1 | - | **51.4** / 0.5 | **51.4** / 0.5 | **51.4** / 0.5 | - |
| CIFAR10 ♠ | ISUN | PBB | **95.6** / 0.1 | **95.6** / 0.1 | 95.4 / 0 | 93.5 / 0 | 15.1 / 0.1 | **13.6** / 0.5 | 16.1 / 0.2 | 30.6 / 0.4 |
| | | TBB | **95.4** / 0.1 | 70 / 0.1 | 95.2 / 0.1 | - | 16.1 / 0.4 | **16** / 0.5 | **16** / 0.4 | - |
| | LSUN (CROP) | PBB | **96.1** / 0.1 | 95.9 / 0.1 | 95.5 / 0.2 | 95.2 / 0.1 | 12.6 / 0.5 | **12.4** / 0.3 | 15.3 / 0.7 | 20.8 / 0.4 |
| | | TBB | **95.2** / 0.1 | 70 / 0.1 | **95.2** / 0.1 | - | 15.8 / 0.7 | 15.8 / 0.7 | **15.7** / 0.7 | - |
| | LSUN (RESIZE) | PBB | **96** / 0 | 95.8 / 0 | 95.7 / 0 | 93.6 / 0 | 13.2 / 0.5 | **13** / 0.2 | 15.2 / 0.4 | 30.3 / 0.4 |
| | | TBB | **95.5** / 0.1 | 70.2 / 0.1 | **95.5** / 0.1 | - | 15.2 / 0.5 | 15.2 / 0.5 | **15.1** / 0.5 | - |
| | TINY (CROP) | PBB | **96** / 0.1 | 95.9 / 0.1 | 95.7 / 0 | 93.6 / 0 | 13.5 / 0.9 | **12.7** / 0.4 | 15.2 / 0.4 | 30.3 / 0.4 |
| | | TBB | 95.5 / 0.1 | 70.3 / 0 | **95.6** / 0 | - | 15.1 / 0.2 | **15** / 0.3 | **15** / 0.2 | - |
| | TINY (RES) | PBB | **95.5** / 0.1 | 95.2 / 0.1 | 95.1 / 0.1 | 93.2 | **14.7** / 0.3 | 14.8 / 0.5 | 17.1 / 0.4 | 31 / 0 |
| | | TBB | **94.9** / 0.1 | 69.7 / 0.1 | 94.9 / 0.1 | - | 16.8 / 0.3 | 16.9 / 0.2 | **16.7** / 0.2 | - |
| CIFAR100 ♠ | ISUN | PBB | **83.3** / 0.1 | 83.1 / 0.1 | 83 / 0.2 | 82.6 / 0.2 | 57.8 / 0.3 | 57.1 / 1 | **56.8** / 0.8 | 60 / 0.4 |
| | | TBB | **82.6** / 0.2 | 79.7 / 0.2 | 82.5 / 0.2 | - | **58.3** / 1 | 58.4 / 1.1 | 58.4 / 1 | - |
| | LSUN (CROP) | PBB | 90.6 / 0 | **90.7** / 0 | 89.9 / 0.1 | 87.5 / 0 | 35.9 / 0.2 | **34.6** / 0.2 | 38.5 / 0.4 | 56.1 / 0.2 |
| | | TBB | **89.4** / 0.1 | 86.2 / 0 | 89 / 0 | - | **39.4** / 0.1 | **39.4** / 0.1 | **39.4** / 0.1 | - |
| | LSUN (RESIZE) | PBB | 83.6 / 0.2 | **83.8** / 0.1 | 83.6 / 0.2 | 83.2 / 0.1 | 55.8 / 0.4 | 54.2 / 0.7 | **54.1** / 0.6 | 59.6 / 0.8 |
| | | TBB | **83.2** / 0.1 | 80.4 / 0.1 | 83.2 / 0.1 | - | 55 / 0.6 | **55** / 0.7 | **55** / 0.6 | - |
| | TINY (CROP) | PBB | 88.3 / 0.1 | **88.5** / 0.1 | 88.1 / 0.1 | 87.7 / 0.1 | 43.2 / 0.5 | **41.5** / 0.7 | 42.9 / 0.4 | 54.3 / 0.1 |
| | | TBB | **87.8** / 0 | 84.7 / 0.2 | 87.5 / 0.1 | - | **43.7** / 0.2 | **43.7** / 0.2 | **43.7** / 0.2 | - |
| | TINY (RES) | PBB | 83.8 / 0.1 | 83.8 / 0.1 | **83.9** / 0.2 | 83 / 0.2 | 57.9 / 0.5 | 56.6 / 0.9 | **55.6** / 1 | 61 / 0.6 |
| | | TBB | **83.6** / 0.1 | 80.7 / 0.1 | 83.5 / 0.1 | - | **55.5** / 0.8 | **55.5** / 0.8 | **55.5** / 0.8 | - |

Table 7: DOCTOR for pure OOD detection. We set : $\epsilon_\alpha = 0$ and $T_\alpha = 15$, $\epsilon_\beta = 0$ and $T_\beta = 1000$, as in [23] for ODIN$_{OOD}$. The baseline results reported below show that DOCTOR is competitive for OOD detection as well since it can reach similar scores or even outperform the baseline.

| DATASET-IN | DATASET-OUT | AUROC % | | | FRR % (95 % TRR) | | |
|---|---|---|---|---|---|---|---|
| | | $D_\alpha$ | $D_\beta$ | ODIN$_{OOD}$ | $D_\alpha$ | $D_\beta$ | ODIN$_{OOD}$ |
| CIFAR10 | ISUN | 98.1 | 97.9 | **98.8** | 8 | 9.1 | **6.3** |
| | TINY (RES) | 97.6 | 97.3 | **98.5** | 9.9 | 11.2 | **7.2** |
| | LSUN (CROP) | **98.6** | 98.2 | 98.2 | **5.4** | 6.9 | 8.7 |
| | TINY (CROP) | 98.9 | 98.5 | **99.1** | 4.6 | 6.4 | **4.3** |

Table 8: Comparison of $D_\alpha$ with ENERGY and ODIN (parameter setting as in [23]) when OOD samples are similar to in-distribution samples.

| DATASET-IN | DATASET-OUT | METHODS | AUROC % | FRR % (95 % TRR) |
|---|---|---|---|---|
| CIFAR100 | CIFAR10 | $D_\alpha$ (PBB) | 76.8 | 64.2 |
| | | ENERGY | 73.3 | 76.4 |
| | | ODIN (OOD) | 70.5 | 79.5 |

Table 9: Comparison of MHLNB (WB) and $D_\alpha$ (PBB).

| DATASET-IN | METHODS | AUROC % | FRR % (95 % TRR) |
|---|---|---|---|
| CIFAR10 | $D_\alpha$ (PBB) | 95.2 | 13.9 |
| | MHLNB (WB) | 49.5 | 97.3 |
| CIFAR100 | $D_\alpha$ (PBB) | 88.2 | 35.7 |
| | MHLNB (WB) | 51.6 | 94.9 |