# OpenReview forum: "DOCTOR: A Simple Method for Detecting Misclassification Errors"
_NeurIPS.cc/2021/Conference — NeurIPS 2021 Spotlight_

### Official Review · Reviewer_EKmb · 2021-07-16

**Rating:** 6
**Confidence:** 3

**Summary:**

Motivated by the tendency of DL models to be overconfident, this paper introduces DOCTOR, a framework for determining when to accept/trust model decisions. The authors derive the tradeoffs of accepting an incorrect classification and rejecting a correct classification to motivate the formulation of DOCTOR, a set of discriminators which takes into account the full softmax values of the model to minimize the occurrence of type 1 and type 2 errors. DOCTOR is able to match or outperform prior works on both in-domain and out-of-domain problems for both totally black box and partially black box models.

**Limitations And Societal Impact:**

The authors did explain that "practitioners who lack the necessary knowledge and skills may incur misinterpretation of the results obtainable with the proposed method", but it would be helpful to go more in depth on how exactly these results could be misinterpreted or under what situations DOCTOR is not suitable for use.

**Main Review:**

I appreciate the authors concern with addressing the issue of whether or not to trust model, as this is incredibly important when applying ML to things like healthcare or finance. The theoretical analysis seems very well thought out and a good motivation for DOCTOR, and the explanations of previous works was very helpful in the experiments. However, the method section is hard to follow due to the sheer amount of notation and minimal verbal intuition; I am still unsure of exactly how the softmax is used within the DOCTOR discriminators. This could be simply because I am not as familiar with this specific line of work, but if the authors could add a bit more intuition similar to what was done in the ODIN paper it would increase the ease of understanding significantly.

My other main concern is the results; it seems like DOCTOR does not achieve significant gains over ODIN or SR, with most gains being <0.5%. Since y'all are using pertained models, is it possible that if these models were trained with a few different initializations that DOCTOR would not outperform the others? Again I think the impact of DOCTOR could be great as it is lightweight and can be applied to many different setups, but I would need more empirical evidence that it is in fact the best choice.

Other nits:
* Why is ODIN not bolded for TinyImagenet in Table 1? It seems to have the same TBB AROC and TBB FRR of D_a, D_b, and SR.



Update: I appreciate the authors clarification on DOCTORs use of the softmax and their explanation as to why as lower FRR while maintaining the same AUROC can be quite useful in practice. Due to this and the fact that DOCTOR has significant theoretical backing and seems relatively easy to use, I have changed my score to a weak reject. Again I want to thank the authors for their thoughtful response!

**Time Spent Reviewing:**

3

---

> ### Author Response · Authors · 2021-08-10
> **We addressed all the points raised by the reviewer.  In particular, we further clarify details of our theoretical contributions in Section 3; we better highlight  that DOCTOR  improves significantly the existent methods (e.g., up to 4% of the false rejection rate (FRR) in the PBB scenario); we discuss some of the limitation of the proposed method.**
>
> We thank you for your valuable comments.
>
> 1. SOFTMAX IN DOCTOR. We will address the potential lack of clarity about the use of the softmax in DOCTOR by including the following explanation: recalling from Basic Definition in Section 2, we interpreted the inferred model $P_{\widehat{Y}|X}\equiv P_{\widehat{Y}|X}(y|x; D_n)$ as the prediction of the class (label) posterior probability given a sample. That is given a sample $x$, $P_{\widehat{Y}=y|X=x} = \text{softmax}(x)_y$,  where $y\in\mathcal{Y}$ and $\mathcal{Y} =$ { $1,\dots, C $ }
>  is the label space. Because of Definition 2 and eq. 10: $D_\alpha (x, \gamma ) = \mathbb{1} \left[ 1 - \widehat{g}(x) > \gamma \cdot \widehat{g}(x) \right] = \mathbb{1} [ 1 -$  $\sum_y$ $\text{softmax}(x)_y^2  > \gamma \cdot \sum_y \text{softmax}(x)_y^2]$, where $y\in\mathcal{Y}$. Obviously, the same reasoning applies to $D_\beta$ because of Definition 2 and eq. 2.
>
>
> 2. DOCTOR w.r.t. THE COMPETITORS. We would like to underline that, while the gains of DOCTOR are not particularly impressive according to the AUROC metric, it is important to emphasize that DOCTOR provides significant improvements, i.e. up to 4% in terms of the false rejection rate (FRR) in the PBB scenario  (see Table 1, Table 2 and Fig. 2). This second metric is the one of practical interest as it gives the percentage of correctly classified samples that are rejected by the detector when the percentage of rejected misclassified samples is high (95 %). A decrease in this amount of 4% can be very significant in practical scenarios. In addition, on the top of these gains, DOCTOR is derived from the theoretical foundations of the optimal discriminator for the problem of detecting misclassification errors.
>
>
> 3. DOCTOR with DIFFERENT INITIALIZATION OF THE MODEL. We think you have raised an interesting point which deserves to be further investigated. In particular, the impact of models trained with different initializations in the performance of DOCTOR while ensuring that the accuracy reached remains unchanged. However, we believe that no significant changes should be expected. The rational intuition for this is due to the invariant properties of the discriminator $D_\alpha$ (see Def. 2) with respect to the soft-probability of the underlying model. Notice that even by permuting the vector of the posterior probabilities, the output of the $g$ function in eq. 10  remains unchanged, as it is a sum of squared values of the softmax probabilities. Of course, we will conduct those simulations to validate our intuition and update the paper accordingly.
>
>
> 4. Thank you for reporting the missing bold letters in the table, we will complete them.
>
>
> 5. LIMITATIONS. Refer to Reviewer Psjq (item #6 of our reply).
>
>
> *We hope we have addressed most of your comments satisfactorily and kindly request you to revise your score.*

---

> ### Author Response · Authors · 2021-08-23
> **We greatly appreciate your careful reading of our rebuttal**
>
> We greatly appreciate your careful reading of our rebuttal and comments which helped us to improve  the quality of our paper. Thank you  for updating consequently your score.  Best regards, the authors.

---

### Official Review · Reviewer_4WST · 2021-07-17

**Rating:** 7
**Confidence:** 3

**Summary:**

The paper proposes a method for detecting misclassified samples in neural networks.
The proposed method aims to approximate to error function of true probability density (which is unknown) of labels given data by using known statistics.
To achieve this, known statistics that bounds the true error function are derived in the paper which are used to define discriminators to determine correctly and incorrectly classified samples.
The first discriminator denoted as D_\alpha uses sum of squared softmax probabilities and the second discriminator is based on the predicted model for the posterior class probability denoted as D_\beta.
The methods is tested in 2 settings: 1) Partially Black Box (PBB) where it is assumed that gradients of the network w.r.t input is available for input perturbation which was shown to be useful for OOD detection in earlier work, 2) Totally Black Box (TBB) where it is assumed that only softmax probabilities are available.
The proposed discriminators are compared with well-known OOD detection methods: ODIN and Mahalanobis on multiple image and text datasets.
The results demonstrate that the proposed framework (DOCTOR) outperforms OOD detection methods on detecting misclassification error.



**Limitations And Societal Impact:**

- Please see weaknesses section about for the limitations

- I don't see any potential negative impact of the work.

**Main Review:**

** Strenghts **

- The proposed discriminators are very simple yet seems very effective based on the results presented in the paper.
- The paper presents statistical derivations of the proposed discriminators which supports why the proposed discriminators work.
- Extensive experimental evaluations on both image and text dataset demonstrate that DOCTOR can detect misclassification error on the dataset they are trained on (see Table 1)
- The paper also presents experiments when OOD samples are given in test time along with the in-distribution samples. In this case, networks can make mistakes on both in-distribution and OOD samples. Experimental results show that DOCTOR can detect misclassified samples in this setting as well. (see Table 2)

** Weaknesses **
- The paper defines 2 settings for the experiments: PBB and TBB. Most of the state-of-the-art OOD detection methods (e.g. Mahalanobis) focus on fully transparent setting where they assume that intermediate layer representations of the networks are available. I understand the motivation behind using PBB and TBB that they are more realistic settings. However, I believe it is crucial to compare the method with the fully transparent setting of the existing algorithms to better position the method within the related literature.

- The experiments of detecting misclassification error in presence of OOD samples are a bit limited. The results are only compared with ODIN in this setting. I would expect to see more comparisons with some of the recent and more powerful methods such as Mahalanobis, Energy-Based OOD detection [1], Generalized ODIN [2].

[1] Liu et al. Energy-based Out-of-distribution Detection
[2] Hsu et al. Generalized ODIN: Detecting Out-of-distribution Image without Learning from Out-of-distribution Data

-  I believe a general problem with the OOD detection literature is that in-distribution and OOD datasets are quite dissimilar and most of the recent OOD detection methods perform quite well on these benchmark. A more challenging problem that needs to be solved is detecting OOD samples when OOD images are quite similar to each other. There are some works focuses on such settings e.g. where CIFAR100 is in-distribution and CIFAR10 is OOD [3] which achieve AUROC around 0.85. I think it is important to perform experiments on such more challenging setting to see the limits of the OOD detection methods.

[3] Zhang et al. Hybrid Models for Open Set Recognition

**Time Spent Reviewing:**

4

---

> ### Author Response · Authors · 2021-08-10
> **We addressed all the points raised by the reviewer. In particular, we incorporate the comparison with Mahalanobis (fully transparent setting) [4] and Energy Score [1] both for the pure misclassification scenario and for misclassification in presence of OOD samples; we evaluate the performance of DOCTOR for OOD detection when  out-distribution and in-distributed samples are similar to each other.**
>
> We thank you very much for your appreciation of our work and your valuable feedback.
>
>
> 1. COMPARISON WITH THE WHITE BOX (WB) SCENARIO: As you correctly underline, the proposed framework is thought to be effective in the realistic case in which a white box access to the model is either not possible or too complex to be implemented. According to your suggestion, we conducted the following experiments and we will update the paper accordingly.
>
> | DATASET  | METHOD                     | AUROC % | FRR % (AT 95% TRR) |
> |----------------|-------------------------------|:---------------:|:------------------------------:|
> | CIFAR10    | $D_\alpha$                 |   95.2          |          13.9                     |
> | CIFAR10    | ENERGY SCORE [1]  |   91.1          |          34.7                     |
> | CIFAR10    | MHLNB (WB) [4]         |   49.5          |          97.3                     |
> | CIFAR100  | $D_\alpha$                 |   88.2           |          35.7                    |
> | CIFAR100  | ENERGY SCORE [1] |   78.7           |          65.4                    |
> | CIFAR100  | MHLNB (WB) [4]        |   51.6           |          94.9                    |
>
> It is worth clarifying the results in the above table to motivate the performance obtained using the Mahalanobis-based discriminator (MHLNB - WB) for the misclassification detection problem and the issues it raises. First of all, we emphasize that given a network and an input sample DOCTOR only needs to access the logits output of the network in order to perform the detection. On the contrary, the detector based on Mahalanobis distance  consists of 3 steps:
> - estimation of the class mean and covariance matrix;
> - features extraction according to the Mahalanobis score function;
> - aggregation of the scores obtained layer by layer in order to obtain a decision a rule for the discriminator.
>
> Clearly, the Mahalanobis distance-based method requires additional samples compared to DOCTOR. Although estimating the mean and the covariance matrix is possible by exploiting samples from the benchmark training set (e.g. CIFAR10, CIFAR100, ...), this method still needs additional (different from training) samples for learning the linear regressor intended to distinguish between correctly (positive) and incorrectly (negative) classified samples. In order to generate the negative samples, we consider the use of adversarial examples generated through Projected Gradient Descent Attack (magnitude of the perturbation 0.0031), which does not assume any knowledge about the test set.
>
>
> 2. MISCLASSIFICATION DETECTION IN PRESENCE OF OOD SAMPLES: We thank you for the suggested references. According to your suggestion, we add several results in order to better position the proposed method within the related literature.
> - MHLNB (WB) [4]: In the beginning, we did not include these simulations for the reasons described in the previous point. In order to use Mahalanobis,  we need to train the regressor on samples drawn from the in-distribution dataset  (in this example CIFAR10) but correctly classified, and samples drawn from the in-distribution but wrongly classified (together with a  percentage of samples drawn from out-distribution). We think that the lack of additional samples (in particular, misclassified samples)  to train the regressor prevents  MHLNB (WB) to exploit the full information across the layers and thus, DOCTOR outperforms MHLNB (WB). This is why in the beginning we have chosen not to compare with OOD methods that require training over misclassified examples.
> - ENERGY SCORE [2]: we include in the table the results of the simulations for the OOD setting and in the misclassification setting. We compared only with the ENERGY SCORE method in [1], as the ENERGY FINE TUNING method in [1] and the method in [5] do not involve the pre-trained network (we highlight that the goal of DOCTOR is to provide a way of measuring the probability of misclassification of a pre-trained classifier and not to provide a way of training a classifier in order to facilitate the detection mechanism). We decided not to add these comparisons in our initial version of the paper because of their poor performance w.r.t. to the results obtained within our framework. In particular, in the most recent Arxiv version of [1],  eq. 4 is used to build a discriminator in the black box scenario and is strictly related to both ODIN and our framework. However, w.r.t. to the methods based on the softmax, the energy score does not retain the information about the maximum probability of the soft-prediction, which appears to be fundamental for detecting misclassification examples.
>
> In the following table, ODIN_OOD denotes the same parameter setting as in [2]. The results are in terms of  *mean / standard deviation.*
>
> |   Dataset-in  | Dataset-out |   AUROC %  |     AUROC %        |       AUROC %     |      AUROC %          |   AUROC %                  |        AUROC %           | FRR % (AT 95% TRR)   | FRR % (AT 95% TRR)             |  FRR % (AT 95% TRR)          |   FRR % (AT 95% TRR)            |    FRR % (AT 95% TRR)                 |      FRR % (AT 95% TRR)              |
> |:-------------:|:-----------:|:----------:|:----------:|:----------:|:--------------:|:-------------------:|:-----------------:|:------------------:|:----------:|:----------:|:-------------:|:-------------------:|:------------------:|
> |    |  |   $D_\alpha$  |   $D_\beta$   |    ODIN    | ODIN (OOD) | ENERGY SCORE [1] | MHLNB (WB) [4] |       $D_\alpha$      |   $D_\beta$   |    ODIN    | ODIN (OOD) | ENERGY SCORE [1] | MHLNB  (WB) [4] |
> |  CIFAR10 ♣ |     iSun    | 95.4 / 0.1 | 95.1 / 0.1 | 94.6 / 0.1 |    89.6 / 0    |      92.4 / 0.1     |     54.5 / 0.1    |      14 / 0.5      | 13.5 / 0.4 | 17.2 / 0.3 |    38.9 / 0   |      32.2 / 0.1     |      92 / 0.1      |
> | CIFAR 10 ♣ |  Tiny (res) | 95.2 / 0.1 |  94.9 / 0  | 94.6 / 0.1 |    89.6 / 0    |      92.3 / 0.1     |      56.2 / 0     |      14 / 0.4      |  14 / 0.5  | 17.8 / 0.4 |    38.9 / 0   |      32.2 / 01      |     90.3 / 0.2     |
> |  CIFAR10 ♦ |     iSun    | 95.5 / 0.1 | 95.3 / 0.1 | 94.9 / 0.1 |    91.5 / 0    |       92.9 / 0      |     54.5 / 0.1    |     14.4 / 0.6     | 13.4 / 0.2 | 16.8 / 0.5 |    34 / 0.1   |        27 / 1       |      92 / 0.2      |
> |  CIFAR10 ♦ |  Tiny (res) | 95.4 / 0.1 |  95 / 0.1  | 94.8 / 0.1 |     91.4 /0    |       92.8 / 0      |     56.2 / 0.1    |      15 / 0.1      | 14.8 / 0.7 |  17 / 0.5  |   34.5 / 0.9  |      28.8 / 1.9     |      90 / 0.3      |
> |  CIFAR10 ♠ |     iSun    | 95.6 / 0.1 |  95.6 / 0  |  95.4 / 0  |    93.5 / 0    |      93.6 / 0.1     |      54.6 / 0     |     15.1 / 0.1     | 13.6 / 0.5 | 16.1 / 0.2 |   30.6 / 0.4  |      25.1 / 0.2     |      92 / 0.2      |
> |  CIFAR10 ♠ |  Tiny (res) | 95.5 / 0.1 | 95.2 / 0.1 | 95.1 / 0.1 |     93.2 /0    |       93.5 / 0      |     56.2 / 0.2    |     14.7 / 0.3     | 14.8 / 0.5 | 17.1 / 0.4 |     31 / 0    |      25.6 / 0.3     |     90.2 / 0.1     |
>
> 3. OOD SAMPLES WHEN OOD IMAGES ARE QUITE SIMILAR TO EACH OTHER: we thank you for bringing us the problem presented in [3] which is of the uttermost importance. We tested DOCTOR in pure OOD setting, considering CIFAR100 as in-distribution and CIFAR10 as out-distribution. The results below show that DOCTOR optimized as in our paper outperforms ODIN (optimized as described in [2]) and ENERGY_SCORE. This is particularly promising as it shows that DOCTOR, without performing any training and without been particularly optimized for OOD detection, can perform well on a wider variety of problems. Of course, this opens the way for further investigation.
>
> |               DATASET               |        METHOD        | AUROC % | FRR %  (AT 95% TRR) |
> |-----------------------------------|--------------------|:-------:|:----------------------:|
> | CIFAR100 (in) + CIFAR10 (out) |        $D_\alpha$       |   76.8  |          64.2          |
> |      CIFAR100 (in) + CIFAR10 (out)                               | ENERGY SCORE [1] |   73.3  |          76.4          |
> |       CIFAR100 (in) + CIFAR10 (out)                              |     ODIN (OOD)    |   70.5  |          79.5          |
>
> *We hope we have addressed most of your comments satisfactorily and kindly request you to revise your score.*
>
> References:
>
> [1] Liu et al. Energy-based Out-of-distribution Detection.
>
> [2] Liang et al. Enhancing The Reliability of Out-of-distribution Image Detection in Neural Networks.
>
> [3] Zhang et al. Hybrid Models for Open Set Recognition
>
> [4] Lee et al. A Simple Unified Framework for Detecting Out-of-Distribution Samples and Adversarial Attacks.
>
> [5] Hsu et al. Generalized ODIN: Detecting Out-of-distribution Image without Learning from Out-of-distribution Data

---

> > ### Comment · Reviewer_4WST · 2021-08-22
> > **I increase my score by 1 to support acceptance of this paper.**
> >
> > I would like to thank to authors for the detailed rebuttal. I read the reviews of other reviewers and the rebuttal. I think authors addressed my of the concerns raised by the reviewers. I increase my score by 1 to support publication of this paper.

---

> > > ### Author Response · Authors · 2021-08-23
> > > **We greatly appreciate the strong support to our work**
> > >
> > > We greatly appreciate your careful reading of our rebuttal and your strong support to our work and comments which helped us to improve significantly the quality of our paper. Best regards, the authors.

---

### Official Review · Reviewer_Psjq · 2021-07-19

**Rating:** 6
**Confidence:** 4

**Summary:**

The paper addresses the important problem of detecting misclassifications in a classification system. The authors consider two scenarios: total black box setting in which only the softmax predictions are available, and a partial black box setting that assumes access to the model parameters. First, the authors derive a bound to approximate the unknown classification error using a combination of known statistics. Based on this, the authors propose a simple method that can detect misclassifications. The method is shown to work well empirically on some benchmark datasets.

**Limitations And Societal Impact:**

I think the authors could include a discussion on the limitations and the failure modes of the method. In addition, if there is a way to combine D_alpha and D_beta into a single method, that would be nice as well.

**Main Review:**

First, in Section 3, the authors derive a bound to approximate the misclassification error using known model statistics. This section was a bit hard to parse, and it took me a while to fully understand. The authors could work on explaining this section better. For instance, the explanation of quantity $\hat{Pe}(x)$ in Equation 2 can be improved. But other than this, this section was interesting to read.

I took a quick glance at the proofs. They seem to be correct, although I did not review them very carefully.

The experiments were conducted well. It was interesting to see that the method consistently improved in most settings compared to softmax response and MHLNB. Overall, I like the results. In all experiments, I feel running them multiple times and reporting aggregate statistics might be beneficial.

Some comments:

Is it possible to combine D_\alpha and D_\beta into a single discriminator?

How to choose which one of D_\alpha or D_\beta to use?

Can the method be used just for OOD detection? I see that authors reported results of misclassification detection in the presence of a fraction of OOD samples in Table 2. But I would be interested to see the performance of just the OOD detection, and compare it with some standard benchmarks.

It would have been good if authors could discuss the failure modes of the approach.


**Time Spent Reviewing:**

3 hours

---

> ### Author Response · Authors · 2021-08-10
> **We addressed all the points raised by the reviewer. In particular, we further clarify details of the theoretical contributions in Section 3; we show results in terms of mean and variance over a set of experiments; we discuss the aggregation and  failure modes for the proposed discriminators; we provide additional simulations for pure OOD detection.**
>
> We thank you for your insightful comments and for the interest in both our theoretical and practical contributions.
>
> 1. Due to space limitations, at the time of the submission we relegated some insightful details on the quantities involved in section 3 to the supplementary material. Notice that $\widehat{Pe}(x)$ represents the probability of misclassification of the sample $x$ with respect to the softmax probability $P_{\widehat{Y}|X}$, which can be interpreted as the model's approximation of the real $P_{Y|X}$, such approximation is good when the model $f$ is well-calibrated and the training samples are a good representation of the real generative distribution. We will add further clarifications and update the definition of $\widehat{Pe}(x)$ in section 3.
>
>
> 2. It is important to distinguish between the results reported in Table 1 and those reported in Table 2. In the first case, since no training procedures or randomness are involved and the experiments are run on popular test sets (e.g. CIFAR10, CIFAR100, ...), repeating the experiments multiple times will not result in different outcomes. On the other hand, in table 2 we report experiments for which in-distributions samples and out-of-distribution samples are concatenated according to different percentages (see lines 320-321 of the paper).
> In this case, in agreement with your intuition, we express the results in terms of mean and variance over 5 iterations, keeping the percentages of in-distribution and out-distribution samples fixed and randomly drawing samples from both categories according to such percentage (in our Github repository we provide the exact data concatenations to facilitate the reproducibility of these results).
>
>
> 3. COMBINATION OF $D_\alpha$ and $D_\beta$. While our current work does not propose any aggregation techniques for the two discriminators, which are considered as standalone solutions for the misclassification detection problem, we acknowledge that the possible aggregation of the two is an interesting subject for future work. Although the problem is open, it may be beneficial to investigate simple aggregation methods in a preliminary phase. For instance, one could try to compute a linear combination of the discriminators' outputs including a weight parameter for each discriminator's contribution (e.g. such a parameter could be tuned to decide which method is given more weight).  However, this would require additional (out of training) samples for the tuning. Another option may be to apply a majority vote system, possibly adding a third framework as a tie-breaker.
>
>
> 4. COMPARISON BETWEEN $D_\alpha$ and $D_\beta$. We observed that $D_\alpha$ is less sensitive to the selection of $T$: for all the datasets, $D_\alpha$ outperforms $D_\beta$ achieving the best AUROCs by setting $T = 1$. Contrary to $D_\alpha$, $D_\beta$ is more sensitive to the value selected for $T$ in the sense that small changes may result in very different values for the measured AUROCs (in Fig. 4 of the Supplementary Material, we reported the comparison of AUROCs obtained via $D_\alpha$ and via $D_\beta$ for different values of T and epsilon). Although both methods achieve almost the same results for adequate choices of $T$ and epsilon values, $D_\alpha$ may be preferable as it requires less tuning than $D_\beta$.
>
>
> 5. DOCTOR FOR PURE OOD DETECTION. It is worth emphasizing that DOCTOR is not targeting OOD detection, which is a rather different problem from the one investigated in this paper. So we did not optimize an ad-hoc input perturbation for DOCTOR within the OOD detection setup, i.e. we kept the same input perturbation proposed for the misclassification detection task. The baseline results reported below show that DOCTOR is competitive for OOD detection as well since it can reach similar scores or even outperform the baseline (e.g., the simulations with LSUN (CROP) show an improvement of the results of 3.3 % in terms of FRR %). We indicate the methods together with their parameter setting. ODIN_OOD denotes the same parameter setting as in Liang et al. Enhancing The Reliability of Out-of-distribution Image Detection in Neural Networks.
>
> | Dataset-in    | Dataset-out    |         AUROC %        |              AUROC %           |    AUROC %           |   FRR % (at 95% TRR)   |  FRR % (at 95% TRR)                        |       FRR % (at 95% TRR)        |
> |:----------------:|:------------------:|:---------------------------:|:-------------------:|:-----------:|:--------------------------------:|:-------------------:|:------------:|
> |                     |                        | $D_\alpha$ $(T=15,$ $\epsilon=0)$ | $D_\beta$ $(T=1000,$ $\epsilon=0)$ | ODIN (OOD) | $D_\alpha$ $(T=15,$ $\epsilon=0)$ | $D_\beta$ $(T=1000,$ $\epsilon=0)$ | ODIN (OOD) |
> |    CIFAR10      |     iSun    |          98.1          |           97.9          |      98.8     |            8           |           9.1           |      6.3      |
> |    CIFAR10  |  Tiny (RES) |          97.6          |           97.3          |      98.5     |           9.9          |           11.2          |      7.2      |
> |    CIFAR10        | LSUN (CROP) |          98.6          |           98.2          |      98.2     |           5.4          |           6.9           |      8.7      |
> |    CIFAR10        | Tiny (CROP) |          98.9          |           98.5          |      99.1     |           4.6          |           6.4           |      4.3      |
>
>
> 6. DISCUSSION ABOUT UTILITY, LIMITATIONS AND FAILURE MODE. We believe that DOCTOR introduces a novel approach within the field of safety AI as it actively focuses on detecting input samples that may potentially result in classification errors. DOCTOR can be exploited by users as a tool to alert in case the risk of a misclassification is high and thus, the user can decide to reject the decision or to take an action accordingly. Notice this can be particularly important in applications of AI to critical systems. It is observed that, in general, SOTA methods for ODD frameworks do not necessarily perform well in predicting misclassification errors. We recognise that some limitations of DOCTOR arise from the fact that it does not exploit information across the various layers but only soft-predictions are used. In practice, the most important obstacle is the calibration of the threshold (given by "gamma") between the desired fault rejection and acceptance rates, which requires additional validation samples. However, quite often, the cost of collecting data for this operation can be prohibitive, making difficult or too expensive to perform such calibration. We will add a detailed discussion about it in the conclusions.
>
> *We hope we have addressed most of your comments satisfactorily and kindly request you to revise your score.*

---

### Decision · Program_Chairs · 2021-09-27

**Decision:**

Accept (Spotlight)

**Comment:**

This submission suggest a new method for detecting out-of-distribution samples for object detection and the reviewers seem to be in agreement that the submission has sufficient novelty and potential impact for an acceptance. Given the detailed new experimental comparisons with other methods, I suggest that paper is accepted for a spotlight presentation at NeurIPS.